# Progress in Electrochemical (Bio)Sensors for Monitoring Wine Production

**Alina Vasilescu** [1],*, **Pablo Fanjul-Bolado** [2], **Ana-Maria Titoiu** [1], **Roxana Porumb** [3] **and Petru Epure** [4]

1    International Centre of Biodynamics, 1B Intrarea Portocalelor, 060101 Bucharest, Romania; tanamaria@biodyn.ro
2    Metrohm Dropsens, S.L., Ed.CEEI, Parque Tecnológico de Asturias, 33428 Llanera, Spain; pablo.fanjul@metrohm.com
3    Research and Development Institute for Vine and Wine, 2 Valea Mantei, Valea Calugareasca, 107620 Prahova, Romania; porumbroxana@yahoo.com
4    Epi Sistem SRL, 145 Brasovului Blv, 505600 Sacele, Brasov, Romania; petru.epure@epi.ro
*    Correspondence: avasilescu@biodyn.ro; Tel.: +40-21-310-4354

**Abstract:** Electrochemical sensors and biosensors have been proposed as fast and cost effective analytical tools, meeting the robustness and performance requirements for industrial process monitoring. In wine production, electrochemical biosensors have proven useful for monitoring critical parameters related to alcoholic fermentation (AF), malolactic fermentation (MLF), determining the impact of the various technological steps and treatments on wine quality, or assessing the differences due to wine age, grape variety, vineyard or geographical region. This review summarizes the current information on the voltamperometric biosensors developed for monitoring wine production with a focus on sensing concepts tested in industry-like settings and on the main quality parameters such as glucose, alcohol, malic and lactic acids, phenolic compounds and allergens. Recent progress featuring nanomaterial-enabled enhancement of sensor performance and applications based on screen-printed electrodes is emphasized. A case study presents the monitoring of alcoholic fermentation based on commercial biosensors adapted with minimal method development for the detection of glucose and phenolic compounds in wine and included in an automated monitoring system. The current challenges and perspectives for the wider application of electrochemical sensors in monitoring industrial processes such as wine production are discussed.

**Keywords:** voltammetry; amperometry; wine; biosensors; fermentation; allergen

## 1. Introduction

Wine production is an important economic sector generating large revenues, e.g., 293 million hectoliters were produced in 2018 worldwide. Wine is a complex mixture of molecules originating from grapes and from the different biochemical processes taking place during the production, from the alcoholic fermentation (AF) to ageing.

Monitoring the progress of wine production and the quality and safety of the wine requires the quantitative determination of a series of composition parameters in addition to temperature, density, acidity, pH or organoleptic characteristics. Besides the main indicators of the alcoholic and malolactic fermentation (MLF, ethanol, glucose, malic acid, lactic acid), phenols and sulphite, knowing the levels of glycerol, gluconic acid, biogenic amines, acetaldehyde, allergens etc. would help winemakers to better control the production process and determine the final taste, quality or flavor of wine.

These composition parameters can be determined by a variety of analytical methods, for example, the official methods established by the International Organization of Vine and Wine (in French

"Organisation Internationale de la Vigne et du Vin", OIV) go from simple measurements of the refractive index to determine the alcoholic degree, through enzymatic colorimetric kits or ion chromatography for organic acids, to spectrophotometric methods for total polyphenols, and HPLC (high-performance liquid chromatography) for biogenic amines etc. Many of these procedures are tedious, require bulky instrumentation, with expensive and time-consuming maintenance protocols, to be used by skilled technicians. As an alternative analytical tool, (bio)sensors with different detection modes promise to provide fast, precise, low cost analysis as well as and portability. Compared to the optical detection mode, electrochemical biosensing provides good sensitivity with some advantages such as the possibility to analyze colored and turbid samples and low-cost analytical equipment. In a convenient and widely used approach, the biosensors can be easily obtained by the combination of an enzyme as recognizing element with a screen-printed electrode as transducer so the final device can be considered disposable for batch analysis or a consumable to be placed in a flow injection analysis system for continuous monitoring. Several enzymes from the groups of oxidases or dehydrogenases can be used for this purpose. In some cases, such as the determination of reducing sugars or total polyphenols, it is also possible to circumvent the need for biorecognition elements by using gold screen-printed electrodes or carbon-based electrodes modified with different nanomaterials [1–3].

Among the electrochemical detection methods, voltammetry and amperometry were the most used with biosensors for wine analysis due to their combined attributes of simplicity, sensitivity, selectivity and low cost. Amperometric detection consisting in polarizing the electrode at a fixed potential and recording the current due to the electrochemical transformation (oxidation or reduction) of the targeted compound is used in FIA (flow injection analysis) detection systems and with enzyme biosensors. In voltammetry, the potential is scanned in a certain range where the transformation of the analyte of interest takes place at the electrode surface. The density of the peak current produced or the charge acquired up to a certain potential value is correlated with the amount of analyte in the sample. Whether the potential is scanned linearly, (as in linear sweep voltammetry, LSV) back and forth between two values (cyclic voltammetry, CV) or in pulses (potential steps) as in differential pulse voltammetry (DPV) and square wave voltammetry (SWV), the result is a current intensity-potential curve that is specific for the analyte. This is due to the fact that the potential at which the electrochemical transformation of a compound occurs is related to its chemical structure and to the nature of the electrode (surface chemistry, roughness, porosity, oxidation degree, material and any modifiers such as nanomaterials, mediators polymers etc.). Obviously, compounds with similar structures are oxidized/reduced at similar potentials on the same electrode and wine is a very complex matrix. However, voltammetry performed at bare or modified electrodes remains a very powerful tool for analyzing wine composition [4]. In particular, applications of cyclic voltammetry include measuring the content of total polyphenols and total antioxidant capacity of wines [5–7], free sulphur dioxide [8,9] or the allergenic protein lysozyme [10]. In addition to CV, DPV [11,12], SWV [13], and LSV [14] have been shown to be very useful for obtaining qualitative and quantitative information regarding the poliphenolic composition [11,13] and the content of sulphites [14] or ascorbic acid [12] in wines.

Voltammetric responses were correlated with organoleptic attributes such as color [15] or astringency [16] and can, moreover, be used to study wine oxidation [8,17] and ageing [18]. More information can be found in several reviews on the determination of wine phenolics [19]; electrochemical methods for determining the antioxidant capacity in food [20], or on electronic tongues (e-tongues) for food and wines [21,22]. Electronic tongues are devices that combine a set of electrochemical sensors with a chemometric method for data interpretation. Combined information from different voltammetric sensors contribute to a unique fingerprint of the sample, which can be used for many applications [21,22] including wine discrimination according to geographic region [23], determination of polyphenols [24], free sulphur dioxide and ascorbic acid [17]; characterization of oxidation patterns in white wines [25] etc.

This review summarizes the main applications of voltamperometric (bio)sensors in wine analysis covering the monitoring of alcoholic and malolactic fermentation, the determination of phenolic compounds and total antioxidant activity, the analysis of allergenic proteins and of sulphite. Relevant examples of biosensors and their main analytical characteristics are summarized in several tables, without the intention of being exhaustive but to deliver an image of the variety of concepts developed so far. Some biosensors are discussed in more detail to emphasize either studies characterized by a more extensive evaluation with wine samples or promising new detection approaches based on nanomaterials. A case study is also included, detailing an automated detection system including amperometric (bio)sensors based on screen-printed electrodes for the detection of glucose and phenolic compounds during the alcoholic fermentation.

## 2. Applications of Voltammeric Biosensors in Wine Production

### 2.1. Biosensors for Monitoring Alcoholic Fermentation

Alcoholic fermentation is the primary biochemical process in wine production, where grape sugars are converted into alcohol, carbon dioxide and different metabolites under the action of yeasts. Considering the critical importance of this step for the quality of the wine produced, most biosensor-based systems that comply with industry requirements such as robustness, precision, accuracy etc. and relying mostly on amperometric detection were developed to monitor parameters such as glucose, ethanol, fructose and glycerol, relevant for alcoholic fermentation. In addition to commercial analyzers (discussed in more detail in [26,27] and in Section 2.5), a whole range of (bio)sensor methods were reported, as summarized in Table 1 and in relevant reviews [28,29].

**Table 1.** Electrochemical biosensors for the determination of glucose and ethanol in wines.

| Target Analyte | Application/ Matrix | Analytical Parameters | Biosensor Configuration | Reference Method | Ref. |
|---|---|---|---|---|---|
| Glucose, ethanol, lactate | 12 wines (red and white; dry and sweet) | LR (glucose): 0.04–2.5 mM, LR (ethanol): 0.3–20 mM, LR (lactate): 0.008–1 mM; Storage stability: >2 months (ethanol, glucose); 4 days (lactate) | Pt printed electrodes/GOx (AOx, LOx) Batch | HPLC | [30] |
| Glucose, Fructose, Ethanol | microalcoholic fermentations 23 red wine samples | LR (glucose): 0.02–0.7 mM LR (fructose): 0.02–0.7 mM, LR (ethanol): 0.05–0.5 mM Stability: 90% after 6 months (glucose), 1 month (ethanol) and 15 days (fructose) Recovery in spiked wines: 95–105% | SPGE/PB/GOx (AOx) (glucose, ethanol) SPGE/PMS/FDH (fructose) Batch. | Spectrophotometric kit | [31] |
| Glucose, ethanol, lactate | Wines, milk, Fermentation media | LR (glucose): 0.05–1.10 mM LR (ethanol): 0.09–0.90 mM LR (lactate): 1–53 mM | Biosensor array SPGE/PB/GOx (AOx, LOx) Batch | HPLC | [32] |
| Glucose | Lab-scale fermentation (7 samples) | LR: 5-200 mg/L LOD:16.2 mg/L LOQ: 54.1 mg/L | H), G6P-DH, NAD$^+$ from commercial kit, detection of NADH with SPE/O-MWCNT Batch | Glucose kit | [33] |
| Glucose, ethanol | Fermentation broth-24 wines | LR: 0.3-7.8 mM LOD: 0.1 mM Stability: 50% after 30 days | GOx/ADH/Fe$_3$O$_4$@Au/ MnO$_2$-CPE Batch | Glucose meter | [34] |

**Table 1.** *Cont.*

| Target Analyte | Application/ Matrix | Analytical Parameters | Biosensor Configuration | Reference Method | Ref. |
|---|---|---|---|---|---|
| Glucose | 10 red and white wines | LR: $10^{-6}$–$10^{-3}$ M | GC/PB/GOx/Nafion Semiautomatic FIA analyzer | Spectrophotometric assay | [35] |
| Glucose | Commercial red/white wine | LR: 5–1000 µM<br>LOD: 20 µM<br>Operational stability: No decrease after 8 h of 55 injection of glucose | SPGE/PB/GOx/TEOS–PVA/Nafion FIA | HPLC | [36] |
| Glucose | 9 wines (red, rose, white, dry and sweet) | LR1: 0.3–2 g.L$^{-1}$<br>LR2: 2–10 g.L$^{-1}$<br>LR3: 10–50 g.L$^{-1}$ | GOx/HRP/Fc/CPE FIA | Enzymatic kit | [37] |
| Glucose | 2 red wines | LR: 0.02–4.5 mM<br>LOD: 0.005 mM<br>Stability: 92% after 25 days | NiO-GR/GCE Batch | HPLC | [1] |
| Glucose | N/A | LR: 10–25 mM<br>No interference from ethanol | SPGE/Au/TiO$_2$ Batch | N/A | [2] |
| Ethanol | Alcoholic fermentation of wines (6 days) | LR: 1–250 µM<br>LOD: 1 µM | QH-ADH + PVI13dmeOs + PEGDGE SPE On-line SIA analyzer OLGA | Enzymatic kit | [38] |
| Malate | MLF of 3 red wines | LR: $1 \times 10^{-7}$–$1 \times 10^{-6}$ M<br>LOD: $6.3 \times 10^{-8}$M<br>Retains 90% of sensitivity after 37 days | Thick film Au/MDH-DP, NAD$^+$/Ppy-HAR Batch | Colorimetry | [39] |
| L-malate, L-lactate | MLF of synthetic wine induced by *Lactobacillus plantarum* CECT 748$^T$ | LR (malate):$5.2 \times 10^{-7}$–$2.0 \times 10^{-5}$ M<br>LR (lactate):$4.2 \times 10^{-7}$–$2.0 \times 10^{-5}$ M<br>LOD (malate): $5.2 \times 10^{-7}$ M<br>LOD (lactate): $4.2 \times 10^{-7}$ M<br>Stability: 90% of sensitivity after 7 days (malate); 91% of sensitivity after 5 days (lactate) | DM/MDH-DP/TTF/MPA-Au (malate) DM/Lox-HRP/TTF/MPA-Au (lactate) Batch | Enzymatic kits | [40] |
| Lactate | MLF of 3 red wines | LR: $1 \times 10^{-6}$–$1 \times 10^{-4}$ M<br>LOD: $5.2 \times 10^{-7}$ M<br>90% of sensitivity after 40 days | Thick film Au/Lox-HRP/PPy Batch | Colorimetry | [41] |
| Lactate | MLF of red wine (11 samples) | LR: (0.005–1 mM;<br>LOD: 0.005 mM<br>Operational stability: 8 h<br>Lifetime:30 days | SPGE/PB/Lox-TEOS (PVA) FIA | Ion chromatography | [42] |
| L-malic acid, L-lactic acid, citric acid | MLF induced by 2 strains of *Oenococcus oeni* (16 samples) | LR (malate): $10^{-5}$–$4 \times 10^{-4}$ M<br>LR (lactate):$5 \times 10^{-6}$–$10^{-3}$ M<br>LOD (malate): $3 \times 10^{-6}$ M<br>LOD(lactate): $2 \times 10^{-6}$ M<br>Stability after 150 injections: 90% of response (malate), 65% of response (lactate) | Pt/LOx on Nylon membrane Pt/ME enzymatic reactor; PMS FIA | Spectrophotometric | [43] |

Abbreviations: GOx—glucose oxidase, AOx-alcohol oxidase, LOx-lactate oxidase, HPLC-high performance liquid chromatography, PB—Prussian Blue, SPGE—graphite screen-printed electrode, PMS—phenazine methansulphate, FDH-fructose dehydrohenase, G6P-DH—glucose-6-phosphate dehydrogenase, NAD+—nicotinamide adenine dinucleotide, NADH-reduced adenine dinucleotide, O-MWCNT—oxidized-multiwalled carbon nanotubes, ADH—alcohol dehydrogenase, Fe$_3$O$_4$@Au—core-shell iron oxide-gold nanoparticles, MnO$_2$-CPE—manganese dioxide modified carbon paste electrode GC—glassy carbon, FIA—flow injection analysis, TEOS—tetraethyl orthosilicate, PVA—polyvinyl alcohol, HRP—horseradish peroxidase, Fc—ferrocene, NiO-GR—nickel(II) oxide and reduced graphene, GA—glutaraldehyde, QH-ADH—quinohemoprotein alcohol dehydrogenase, PVI13dmeOs—Os-complexed poly(1-vinylimidazole) redox polymer, PEDGE: poly(ethylene glycol) diglycidyl ether, OLGA—On-line General SIA analyzer, MDH—malate dehydrogenase, DP—diaphorase, PPY—polypyrrole, HAR—hexaammineruthenium (III) chloride, DM—dialysis membrane, TTF—tetrathiafulvalene, MPA—self-assembled monolayer of mercaptopropionic acid, ME-malic enzyme.

The (bio)sensors presented in Table 1 range from one-shot disposable devices measuring glucose to systems able to monitor in parallel several composition parameters, all having a response time

of maximum 3 min and promising analytical characteristics, adequate for application in alcoholic fermentation monitoring. Some biosensors still await verification of their performances on larger sets of wine samples and in real industrial settings.

The detection of glucose recorded significant progress since the first glucose analyzer introduced on the market by Yellow Springs Instruments (USA) in 1975 and still remains the most successful commercial application of biosensors, representing 85% of the global biosensor market. By far the major application remains in the biomedical field, however advances in this field from single-use tests to the continuous monitoring of glucose [44] prompted also the progress in food-related applications, for monitoring industrial fermentation processes. Glucose biosensors were extensively reviewed [44–46], the majority of electrochemical devices relying on glucose oxidase (GOx) or glucose dehydrogenase for specific detection. The non-enzymatic detection of glucose is a more recent alternative, based on carbon, Au or indium tin oxide (ITO) electrodes that are modified with multiwalled carbon nanotubes (MWCNT), Pd, Pt, Au, transition metals and metal oxides etc. [47].

Conveniently, commercial glucose biosensors developed for blood analysis could be used also for the quantitative determination of glucose during fermentation processes, with appropriate calibration and sample dilution, e.g., as demonstrated in a 1995 work featuring an Exactech blood glucose sensor for determining glucose in fermentation media [48]. Glucose oxidase-based screen-printed electrodes are available from several vendors including Dropsens-Metrohm (Spain), BVT Technologies (Czech Republic), Gwent Group (UK) etc. Additionally, Dropsens-Metrohm offers NiO-modified carbon electrodes and Au electrodes that could potentially be used for sugars monitoring in fermentation processes. Some adaptations will be required in order to use these sensors for multiple measurements as they were developed as disposable devices.

With regard to the automated, high-throughput analysis of glucose or other important parameters to be monitored during fermentation processes (ethanol, glycerol, lactate etc.), several groups have developed biosensor-based analyzers working in FIA or sequential injection analysis (SIA) modes that reached commercialization, e.g., the SIRE biosensors [49], the On Line General Analyzer (OLGA) [50] etc. The enzyme was provided in liquid form in a compartment separated from the sample by membranes (with fresh enzyme supplied for each test [49] or was immobilized on membranes placed in the vicinity of the amperometric detector [50]. Frequent calibrations have to be performed to account for the decrease in sensor sensitivity over time (e.g., due to membrane clogging, adsorption of components from the fermentation media on the membranes and tubes etc.). The liquid handling part of these analyzers is able of adjusting the sample dilution factors to match the whole range of concentration during the fermentation with the measuring range of the biosensor. These analyzers provide a generic technology where simply changing the enzyme solution or the enzyme membrane and adapting the operational conditions enables to determine other parameters related to wine production (e.g., ethanol, glycerol, lactate etc.), extending the applicability beyond the monitoring of alcoholic fermentation.

For example, detailed fermentation profiles were acquired with an amperometric multi-biosensor FIA system measuring glucose, fructose, ethanol and glycerol at-line, in the winery, for wines obtained by different technological processes [51]. The must was sampled from the fermentor, diluted at least 1:200 and analyzed with the multi-biosensor system. Glucose, fructose, ethanol and glycerol were measured in parallel with biosensors based on membranes with immobilized GOx, fructose dehydrogenase (FDH), alcohol oxidase (AOx) and glycerokinase/glycerol-3-phosphate oxidase, respectively, placed in contact with a platinum electrode. The enzyme membrane was sandwiched between a cellulose acetate membrane and a polyurethane membrane with roles in eliminating interfering compounds and preventing the fouling of the biosensor. Biosensor accuracy was confirmed by the agreement between the measurements performed with the biosensors with those provided by the standard spectrophotometric methods and by performing recovery studies with fortified wines, where recovery factors of 93%–98% and RSDs (relative standard deviations) of 4% were obtained. The simultaneous detection of glucose, fructose, ethanol and glycerol and the kinetics of the fermentation allow verifying the performance of the yeast and to alert on stuck fermentations and on the grapes

infection by *Botrytis cinerea*. The fructose/glucose ratio is an indicator of stuck fermentations due to the different conversion rate of the two sugars by the yeast: glucose is preferentially consumed during the intermediate, tumultuous fermentation phase, and then the remaining sugar (fructose) is metabolized. In a typical fermentation, the fructose:glucose ratio reaches a maximum then decreases, while in a stuck fermentation the residual fructose is not metabolized by the yeast and the ratio fructose: glucose remains important. High glycerol content in wine is an indicator of grape fungal infection by *Botrytis cinerea*. The study of Esti et al. [51] appears as one of the model works that emphasize the applicability of biosensors for controlling the alcoholic fermentation, starting from the multiparameter evaluation with 4 biosensors, to the validation of the biosensors and their large scale evaluation in five fermentation processes conducted at three wineries, using different grape varieties and two maceration techniques, namely pumping over and delestage.

Screen-printed electrodes with surface immobilized enzymes provide low cost, convenient transducers as alternatives to the use of enzyme membranes. An automated system including dedicated measuring units based on screen-printed enzyme biosensors was used to measure in parallel glucose, ethanol, lactate and phenols. The biosensors were obtained by immobilizing glucose oxidase, alcohol oxidase, lactate oxidase and horseradish peroxidase by cross-linking on the surface of carbon electrodes. The automated system featured a liquid handling part that enables sample dilution factors from 1:5 to 1:5000. The linear ranges of the biosensors were 0.05–1 mM (glucose), 0.1–1 mM (ethanol), 0.025–1 mM (lactate) and 0.01–0.05 mM (gallic acid) and the biosensor lifetime ranges from 7 days for the ethanol biosensor to 10 months for the glucose biosensor [52].

A reagentless and stable biosensor working in FIA mode was obtained by immobilizing GOx, HRP and the mediator ferrocene (Fc) in carbon paste [37]. The bi-enzymatic sensor showed no decrease in performance after 50 successive measurements of glucose and had moreover an excellent storage stability, i.e., no decrease in performance was detected after storage for 6 months at 4 °C [37].

Among the few non-enzymatic sensors investigated for the determination of glucose towards the monitoring of fermentation (Table 1), Zhu et al. described a sensor made by modifying a GCE with graphene and NiO. The sensor was able to measure glucose without interference from sucrose, glucose, citric acid, acetic acid, or ethanol and the analysis of 2 red wines with the sensor was in agreement with results obtained for the same samples by an HPLC method. However, the sensor operation requires alkaline electrolyte (0.1 M KOH), moreover this 2013 report is yet to be followed by a more detailed investigation on a larger set of samples.

## 2.2. Biosensors for Monitoring the Malolactic Fermentation

The malolactic fermentation of wines consists in the conversion of L-malic acid into L-lactic acid by the lactic bacteria in wine, with the net result of a decrease in acidity, further reflected in the organoleptic characteristics and the microbiological stability of wines. It was traditionally monitored in wineries by measuring the total acidity or by the semi-quantitative evaluation of the malic and lactic acids via paper chromatography. OIV has defined standard enzymatic spectrophotometric methods for L-malic and L-lactic acid [53]. Besides the corresponding spectrophotometric enzyme kits sold by many vendors, other kits are also available commercially, including colorimetric test strips (the Accuvin™ system from Accuvin LLC, USA) and a system measuring the pressure due to the $CO_2$ produced during the malolactic fermentation (the SC-50 MLF analyzer-kit from Vinmetrica, USA). More recently, wireless systems including pH and temperature sensors installed in barrels were proposed as a new approach allowing continuous and care-free monitoring [54]. As alternatives to all these methods, enzymatic biosensors provide fast (i.e., in minutes) and quantitative specific determination of L-malic and L-lactic acid. Both mono and bi-enzymatic amperometric biosensors have been proposed for the determination of L-malic and L-lactic acids (Table 1). The higher complexity and cost of bi-enzymatic sensors, compared with devices based on a single enzyme, are balanced by a significantly higher sensitivity.

Various sensors have been proposed for the determination of malic acid, where the specific conversion of malic acid was achieved via nicotinamide adeninde dinucleotide (NAD+)-dependent malate dehydrogenase (MDH, EC 1.1.1.37) [40], nicotinamide adeninde dinucleotide phosphate (NADP+)-dependent malic enzyme (ME, EC 1.1.1.40) [43] or a flavin adenine dinucleotide (FAD)-dependent malate quinone oxidoreductase (MQO, EC1.1.99.16) [55]. Lactate oxidase was the preferred catalytic element in biosensors for the determination of lactic acid.

An integrated system for monitoring the malolactic fermentation of wines was described by Gamella et al. [40], including two bi-enzymatic sensors for the determination of malic and lactic acid. To make the biosensors, two enzymes and an electrochemical mediator, tetrathiafulvalene (TTF), were immobilized on an Au disc coated with a self-assembled monolayer (SAM) of mercapto-propionic acid (MPA). The principle of the biosensors and the variation of malic and acid lactic concentration measured during the MLF of a synthetic wine are illustrated in Figure 1.

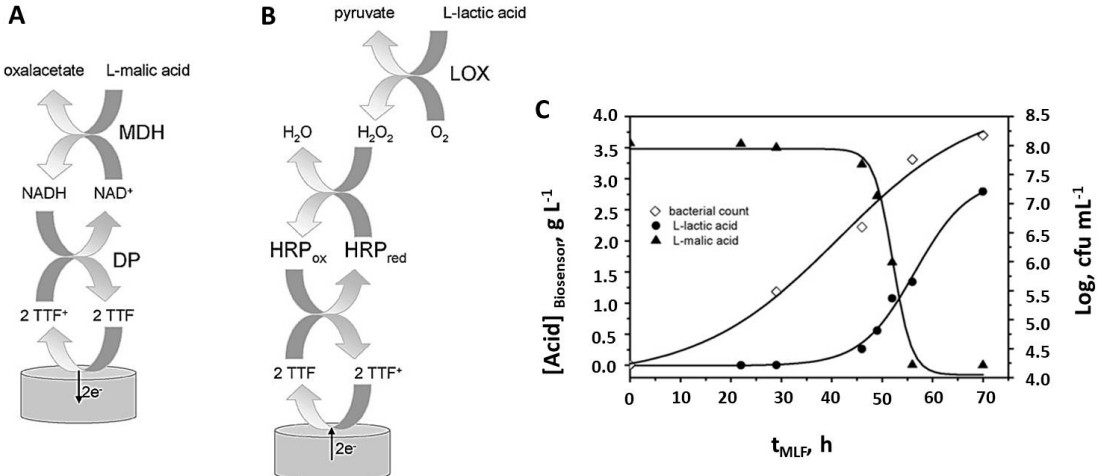

**Figure 1.** Principles of biosensors for malic acid (**A**) and lactic acid (**B**). The biosensor for L-malic acid is based on the cascade reactions catalysed by malate dehydrogenase (MDH) and diaphorase (DP), while the biosensor for L-lactic acid employs lactate oxidase and horseradish peroxidase. Both sensors use TTF as electrochemical mediator. (**C**) The progress of malolactic fermentation of synthetic wines induced by *L. plantarum* CECT 748[T] and monitored with the biosensors for L-malic acid and L-lactic acid. Also shown is the variation of the bacterial count. Reproduced from [40] with permission from Elsevier.

In the reaction catalyzed by MDH, malic acid is converted to oxaloacetate with simultaneous reduction of the enzymatic cofactor NAD+ to NADH. The second enzyme, diaphorase catalyzes the reaction between NADH and the mediator TTF, leading to the regeneration of NAD+. The reduced TTF is finally reoxidized at the electrode surface polarized at an appropriate potential (+0.1 V vs. Ag/AgCl) and the anodic current is proportional with the concentration of malic acid in the sample (Figure 1A). For the determination of lactic acid, the hydrogen peroxide produced in the reaction catalyzed by LOx is reduced to water in the presence of TTF, in a reaction catalyzed by HRP. The oxidized TTF formed as a result is finally reduced on the electrode polarized at −0.05 V and the cathodic current is proportional to the level of lactic acid in the sample (Figure 1). The biosensors were investigated with respect to potential interfering compounds such as ethanol, glycerol, sugars (glucose, fructose, galactose, arabinose) and organic acids (tartric, citric, gluconic, ascorbic, acetic, D-lactic). The only interfering compound was ascorbic acid, however the authors noted that the level of ascorbic acid in wines is much smaller (up to ~0.012 gL$^{-1}$) compared to that of L-malic and L-lactic acids (see Figure 1B). Consequently, the risk of inaccurate measurements due to ascorbic acid is small.

The usefulness of the biosensors was illustrated through their application in the MLF of a synthetic wine, where they allowed to measure accurately L-malic and L-lactic acids and to observe the kinetics

of the MLF. Since the kinetics also depend on the bacterial strain used (besides pH, temperature etc.), there is a good application potential for the biosensors for studying new bacterial strains as starters for MLF.

### 2.3. Biosensors for Phenolic Compounds and Antioxidant Capacity

Phenolic compounds contribute to wine quality, its organoleptic characteristics (e.g., color, astringency, bitterness) as well as to the antioxidant capacity and wine stability [56]. The main groups of phenolic compounds in wines are flavonoids (flavones, flavonols, flavanones, flavanols and anthocyanins) and non-flavonoids (phenolic acids such as vanillic, syringic and gallic acid, hydroxycinnamic acids such as caffeic acid, ferrulic acid, sinapic acid and stilbenes such as resveratrol) [56], with the structures illustrated in Figure 2.

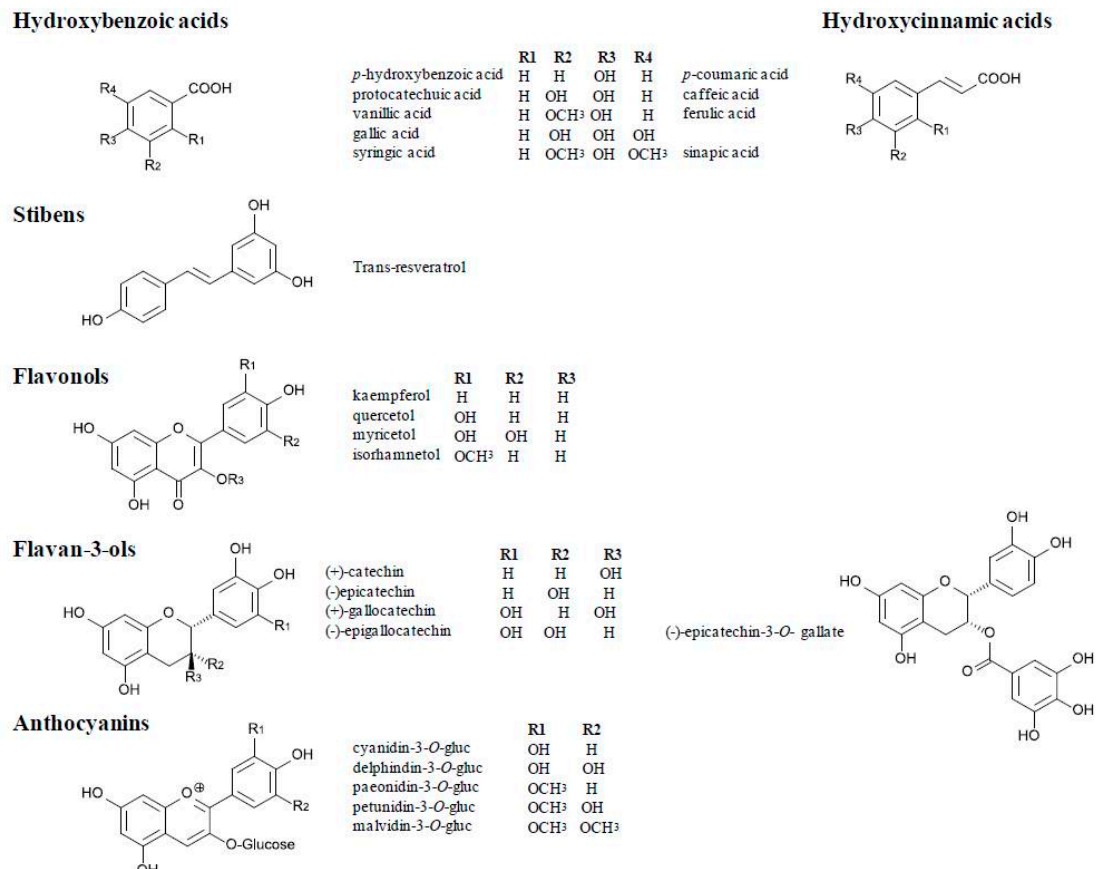

**Figure 2.** The main monomeric phenolic compounds in wines. Reproduced from [56], under Creative Commons Attribution Licence.

Typically these compounds are determined by spectrophotometric and chromatographic procedures [53,56] targeting either the total polyphenols in wines (e.g., the Folin Ciocalteu test and the phenolic index $I_{280}$ representing the absorbance of diluted wine at 280 nm) or individual compounds describing the wine's composition profile. Details of the different methods, including nanomaterial-based approaches and electrochemical (bio)sensors for the determination of phenolic compounds and antioxidant capacity in wine and other food matrices can be found in a series of reviews [19,57–59]. Voltamperommetric methods provide simple, fast and cost-effective quantitative assessment of phenolic compounds and the total antioxidants activity in wine and winery by-products [6,57,59,60] being a viable and attractive alternative to the optical methods [56], which are more time consuming, involving more sophisticated procedures and required clear sample solutions.

Progress in the analysis of wine polyphenols, the advantages brought by electrochemistry and the increasing role of nanomaterials in the (bio)sensing of phenolic compounds and assessment of the total antioxidant capacity in food were recently reviewed [4,56,57,61]. While various gold nanoparticles (AuNPs), AgNPs, metal oxide (ceria, iron oxide) NPs and C-based NPs were used for the detection of antioxidants, the number of electrochemical applications developed so far is sensibly smaller compared to the colorimetric ones [61].

Nanomaterials have (i) improved the sensitivity of electrochemical detection by enabling a higher surface area (hence also higher loading with enzymes) and a better conductivity, (ii) had electrocatalytic effects manifested as a shift in the anodic peak potentials of phenolic compounds towards lower overpotentials, (iii) facilitated an increased selectivity by shifting the potentials required for the oxidation of phenolic compounds and interfering compounds, thus allowing their respective signals to be resolved by voltammetry, (iv) protected the electrode surface against passivation, (v) provided oxidase-like activity resulting in unique interactions between wine polyphenols and the modified electrode. There are infinite possibilities to functionalize the nanomaterials, to combine them with polymers or produce nanocomposite materials, to use materials with different degrees of oxidation or doping, different shapes and sizes in order to modulate the characteristics of the electrochemical sensor towards the desired attributes for the voltamperommetric evaluation of polyphenol content and antioxidant capacity [57].

Without being exhaustive, Table 2 summarizes some of the applications of (bio)sensors relying on voltamperometric procedures for the detection of phenolic compounds and assessment of total antioxidant capacity in wines that include a comparison with standard or more classic approaches. More details and specific examples that include a more extensive testing of wine samples or monitoring different steps in the production process are provided below.

**Table 2.** Voltamperometric (bio)sensors for the determination of phenolic compounds and antioxidant capacity of wines.

| Work Aim | Electrochemical Technique/ Conditions | Sensor Design Details | Principle | Performance Characteristics | Ref. |
|---|---|---|---|---|---|
| TP estimation | FIA-amperometric | GCE-MWCNTs | Polyphenols oxidation. | Phenolic acids: LR: $1.0 \times 10^{-7}$ $1 \times 10^{-4}$ mol L$^{-1}$ | [3] |
| TP | DPV, 0.1 M sodium acetate–acetic acid buffer pH 3.6 | GCE | Polyphenol oxidation | Catechin LR: 1–15 mg.L$^{-1}$ LOD: 0.53 mg.L$^{-1}$ catechin | [11] |
| TP index (as gallic acid) | FIA-Amperometry ($-0.1$ V vs. Ag/AgCl) 0.1 mol L$^{-1}$ Britton–Robinson buffer, pH 5 | SWCNT/MWNCT; TvL or ThL immobilized by PAP cross-linking | Detection of phenols derived quinones | Gallic Acid: LR: 0.1–17.0 mgL$^{-1}$ LOD: 0.1 mgL$^{-1}$ | [62] |
| TP evaluation | FIA-Amperometry ($-100$ mV vs. Ag/AgCl) Buffer: acetate 0.1 M, pH 4.5 | Au-SAM/AuNPs-Linker/Fullerenols/ TvL | Detection of phenols derived quinones | Gallic acid: LR: $3.0 \times 10^{-5}$–$3.0 \times 10^{-4}$ mol L$^{-1}$; LOD: $6.0 \times 10^{-6}$ mol L$^{-1}$ | [63] |
| TP evaluation | Chronoamperometry (50 mV vs. Ag) 0.1 M acetate buffer with 0.1 M KCl pH 5 | GRQDs-MoS2/ nanoflakes; TvL immobilized by electrostatic interaction | Detection of phenols derived quinones | Caffeic acid: LR: $3.8 \times 10^{-7}$–$1.0 \times 10^{-4}$ mol L$^{-1}$ LOD: $3.2 \times 10^{-7}$ mol.L$^{-1}$ | [64] |
| TAC (as gallic acid) | DPV 0.35 V vs. Ag/AgCl; 0.1 mol L$^{-1}$ phosphate buffer pH 2.5 | GCE-SWCNTs | Polyphenols oxidation | Gallic acid LR: $5.0 \times 10^{-7}$ to $1.5 \times 10^{-5}$ mol L$^{-1}$ LOD: $3.0 \times 10^{-7}$ mol L$^{-1}$ | [65] |
| TAC | DPV | GCE-GR reduced-Fe$_2$O$_3$/ Chit | Polyphenol oxidation | Gallic acid: LR: $1.0 \times 10^{-6}$–$1.0 \times 10^{-4}$ mol L$^{-1}$ LOD:$1.5 \times 10^{-7}$ mol L$^{-1}$ | [66] |

Table 2. *Cont.*

| Work Aim | Electrochemical Technique/ Conditions | Sensor Design Details | Principle | Performance Characteristics | Ref. |
|---|---|---|---|---|---|
| TAC | Amperometry at $-0.1$ V vs. Ag/AgCl | SPCE-ceria NPs | Nanoceria mediated polyphenols oxidation to quinones and quinones electrochemical reduction | Gallic acid: $2.0 \times 10^{-6} – 2.0 \times 10^{-5}$ mol $L^{-1}$; LOD: $1.5 \times 10^{-6}$ mol $L^{-1}$; Caffeic acid: LR: $5 \times 10^{-5} – 2 \times 10^{-4}$ mol $L^{-1}$; LOD: $1.5 \times 10^{-5}$ mol$L^{-1}$ Quercetin: LR: $2 \times 10^{-5} – 2 \times 10^{-4}$ mol$L^{-1}$; LOD: $8.6 \times 10^{-6}$ mol$L^{-1}$ Ascorbic acid: LR: $5 \times 10^{-7} – 2 \times 10^{-5}$ mol $L^{-1}$; LOD: $4 \times 10^{-7}$ mol $L^{-1}$ | [67] |
| Catechol, caffeic acid and catechin | CV | Cu NPs/epoxy–graphite-enzyme (tyrosinase, laccase) bioelectronics array | CV and data interpretation by artificial neural network | Average recoveries of 104% (catechol), 117% (caffeic acid) and 122% (catechin) | [68] |
| TAC | Amperometry, FIA system at $-0.100$ V a 1:1 mixture of phosphate buffer pH 6 and ethanol | SPAuE | DPPHC electrochemical reduction of DPPH• | Trolo× LR: $2 \times 10^{-6} – 3 \times 10^{-5}$ mol $L^{-1}$; LOD: $4.5 \; 10^{-7}$ mol $L^{-1}$. Sensitivity: 20.1 µA Lcm$^{-2}$ µmol | [69] |
| TP based on gallic acid | Chronoamperometryat $+0.45$ V 0.5 mol $L^{-1}$ KCl, pH 5 | CPME/Ruthenium oxo-complex | Polyphenol oxidation | Gallic acid LR: 1.12–32.5 mg $L^{-1}$ LOD: 0.08 mg $L^{-1}$ More than 100 measurements RSD < 5.0%, (n = 10) | [70] |
| TP Polyphenols index | Batch-Amperometry, $-0.1$ V; 0.1 M phosphate buffer, pH 7.4 | Tyr-nAu-GCE | Reduction of quinones formed in the enzymatic reaction | Caffeic acid LR: $2.5 \; 10^{-5} – 9.0 \; 10^{-5}$ mol $L^{-1}$ LSensitivity: 82 µA/mM Stability: 18 days | [71] |
| Polyphenol index | FIA, $-0.1$ V Amperometry 0.1 mol $L^{-1}$ citrate buffer of pH 5 | GCE/TvL | Reduction of quinones formed in the enzymatic reaction | Gallic acid LR: 0.04–2.0 mg $L^{-1}$ LOD: 0.04 0.001 mg $L^{-1}$ Caffeic acid LR: 0.001–0.100 mg $L^{-1}$ LOD: 0.001 mg $L^{-1}$ | [72] |
| TP index (as (+) Catechin | FIA-Amperometry $+0.8$ V buffer pH 7.5 | GCE | Polyphenol oxidation | LR: 1-16 mg $L^{-1}$ LOD: 0.30 mg $L^{-1}$ LOQ: 0.99 mg $L^{-1}$ | [73] |
| Gallic acid | DPV phosphate buffer pH 5.8 | SPCE | Polyphenol oxidation | Gallic acid LR: 0.1–2.0 mM LOD: 33 µM | [74] |
| Gallic acid | DPV Supporting electrolyte: 0.1 nitric acid and 0.1 M sulfuric acid | TNrGO-modified GC electrode WCrGO-modified GC | Polyphenol oxidation | Gallic acid: TNrGO-GCE LR: 4.5–76 µM LOD: 1.1 µM WCrGO-GCE LR: 10–100 µM LOD: 3.1 µM | [75] |

**Table 2.** *Cont.*

| Work Aim | Electrochemical Technique/ Conditions | Sensor Design Details | Principle | Performance Characteristics | Ref. |
|---|---|---|---|---|---|
| TAC (as gallic acid) | DPV; 0.1 mol L$^{-1}$ phosphate buffer pH 7.0. | GC modified with Printex L6 nano-carbon and AgNPs | Polyphenol oxidation | Gallic acid: LR: $5.0 \times 10^{-7}$–$8.5 \times 10^{-6}$ mol L$^{-1}$, LOD: $6.63 \times 10^{-8}$ mol L$^{-1}$ Stability: 50 tests | [76] |
| Caffeic acid | DPV 0.4 mol L$^{-1}$ sulfuric acid | Au/MIS made from TEOS; PTEOS; APTMS | Polyphenol oxidation | Caffeic acid LR: 0.15–60.0 μmol L$^{-1}$ LOD: 0.15 μmol L$^{-1}$ Stability: RSD = 3.2% (n = 30) Storage: no significant change after 70 days at room temperature | [77] |
| Caffeic acid | DPV 0.1 M Britton-Robinson buffer pH 2.65 | F-GO/GCE | Polyphenol oxidation | Caffeic Acid: LR: 0.5–100 μmol L$^{-1}$ LOD: 0.18 μmol L$^{-1}$ Stability: 94.7% of response after 30 tests Storage: 95% of activity after 10 days | [78] |
| Caffeic acid | DPV 0.05 M PB solution pH 7 | nitrogen doped carbon modified glassy carbon electrode (NDC/GCE) | Polyphenol oxidation | Caffeic acid: LR: 0.010–350 μmol L$^{-1}$ LOD: 2.4 nmol L$^{-1}$ Stability: 93% of the initial response after 20 test Storage: 93.5% of response 6 weeks of storage | [79] |

Abbreviations: TP—total polyphenols, GCE—glassy carbon electrode, MWCNT—multiwalled carbon nanotubes, SWCNT—single-walled carbon nanotube, TvL, ThL- laccases from *Trametes versicolor* (TvL) and *Trametes hirsuta* (ThL); PAP—polyazetidine prepolimer, SAM-self-assembled monolayer, AuNPs—gold nanoparticles, GRQDs—graphene quantum dots, TAC: total antioxidant capacity, Chit-chitosan, CuNP-copper nanoparticles, SPAuE—screen printed Au electrode, CPME—carbon paste modified electrode, Tyr—tyrosinase, nAu-GCE—glassy carbon electrode modified with electrodeposited gold nanoparticles, TNrGO/WCrGO—titanium nitride or wolframcarbide-doped reduced graphene oxide, MIS—molecularly imprinted siloxanes, PTEOS—phenyltri-ethoxysilane, TEOS—tetraethoxysilane, PTMS—3-aminopropyltrimethoxysilane, F-GO—fluorine-doped graphene oxide.

## 2.3.1. Biosensors for the Quantitative Determination of Phenolic Compounds

The quantitative voltamperometric determination of phenolic compounds was achieved by: (1) the direct oxidation of wines on bare or chemically/nanomaterial modified electrodes, (2) enzyme catalyzed transformation of phenolic compounds (by tyrosinase, laccase or peroxidase) and amperometric detection of the reaction products, or consumed $O_2/H_2O_2$ and (3) oxidation on electrodes modified with nanoparticles with biomimetic (oxidase-like) activity. While the vast majority of these (bio)sensors were aimed at measuring the total content of phenolic compounds in wine, some methods targeted the detection of specific phenolic compounds, such as caffeic acid or gallic acid. Gallic acid is considered the main phenolic acid in wines and is used as a reference for expressing both the content of total phenolic compounds and the total antioxidant capacity. Among the four major representatives of hydroxycinamic acid in wines, (including also p-coumaric, ferulic and sinapic acids) caffeic acid is the most abundant in white wines. The electrochemistry of phenolic compounds from wines was extensively studied [6,80]. As emphasized by these studies, the selective detection of individual phenolic compounds is challenging due to their close oxidation potentials on solid electrodes, e.g., catechin, caffeic acid, quercetin and gallic acid for example are oxidized around 0.38–0.45 V on glassy carbon electrodes [6]. In addition, when using enzymatic sensors, the rates of biocatalytic conversion by laccase of many phenols in musts and wine are similar [81]. In these conditions, approaches towards selective detection relied either on (i) the higher abundance in wine combined with enhanced sensitivity of the electrochemical detection of the targeted phenolic compound compared to similar molecules, (ii) the use of electrode modifiers (nanomaterials, molecularly imprinted polymers-MIPs, conducting polymers, mediators) or (iii) a set of several different (bio)sensors (such as in e-tongues) allowing

unique electrochemical detection patterns to be recorded and discrimination among similar molecules. Typically, diluting the wine with the supporting electrolyte was the only pre-treatment necessary before the electrochemical measurements. In most studies the checking of potential interference focused, besides other phenolic compounds, on ethanol, sugars (mainly glucose), sulphite and ascorbic acid. While sulphite and ascorbic interfered in the detection of phenolic compounds by direct oxidation on carbon electrodes, solutions to circumvent their effect and increase the accuracy of the measurements were found as detailed below.

Direct Oxidation of Phenolic Compounds from Wines on Bare or
Chemically/Nanomaterial-Modified Electrodes

The electroactivity of phenolic compounds from wine make it possible to determine their concentration based on the current intensity recorded at the potential values specific for their electrochemical transformations. Due to similar chemical structures, many phenolic compounds are oxidized in the same potential window on the surface of carbon electrodes, moreover, ascorbic acid and sulfites may also interfere [82]. While the determination of the total polyphenols content (the "polyphenol index") remains the major application related to wine production of the direct electrochemical oxidation methods, a series of voltammetry approaches were aimed at measuring the levels of individual phenolic compounds (e.g., caffeic acid) or at the simultaneous detection of other parameters relevant for wine quality such as sulfites, ascorbic acid and tartaric acid [82,83]. For this purpose, different degrees of selectivity were achieved by modifying the electrode surface with polymers such as poly(3,4-ethylenedioxythiophene) (PEDOT) [82], with nanomaterials [79,84] or molecularly imprinted polymers [77].

The voltammetric determination of wine polyphenols has evolved considerably in the past 20 years. In 1998, after comparing the chromatograms with amperometric detection at +0.4 V and +0.8 V recorded for red and white wines Mannino et al. [85] proposed a test for estimating the "real antioxidant power". The test was based on a FIA system, using a glassy carbon electrode (GCE) polarized at +0.4 V that takes into account the easily oxidized phenolic compounds in wine, i.e., with higher antioxidant power. These include gallic acid, caffeic acid and its derivatives, catechins (in red and white wines) as well as flavonols (e.g., rutin, myricetin, quercetin kaempferol and iso-rhamnetin) found in red wines [85].

Later on, Kilmartin et al. [6–8], Oliveira Brett [80] and Seruga et al. [11] investigated in more detail the oxidation of wine polyphenols on solid carbon electrodes by CV and DPV. A typical voltammogram for a red wine obtained at a classic GCE features 3 anodic peaks (Figure 3A,B) while the voltammograms of white wines only show 2 peaks. The peak at the lowest potential, i.e., at around +0.4 V (P1) is due to the oxidation of phenolic compounds that contain o-dihydroxybenzene functionalities, including hydroxycinnamic acids such as caffeic acid in white wines, phenolic acids such as gallic acid and catechin-type flavonoids (i.e., with o-diphenol groups in the B ring, see Figure 2) in the red wines [8]. The second peak (P2), mainly observed in red wines at around +0.65 V is due to malvidin anthocyanins with contributions also from less easily oxidized phenolic compounds such as trans-resveratrol (a stilbene) and ferulic acid (a phenolic acid) [6]. Finally, P3 at +0.8–+1.0 V is attributed to the oxidation of the hydroxyl group at position 3 on the C-ring of catechin-type flavonoids in red wines, to less oxidisable phenolic acids (p-coumaric and vanillic acids) [6] and larger polymeric phenolic compounds in white wines [8]. Similar characteristics were revealed by SWV [13] (Figure 3C). The specific features of the voltammogram, i.e., peak heights and peak potentials are related to the composition of wines, where in addition to polyphenols, sulphite and ascorbic acid contributed also to the recorded current intensities [86].

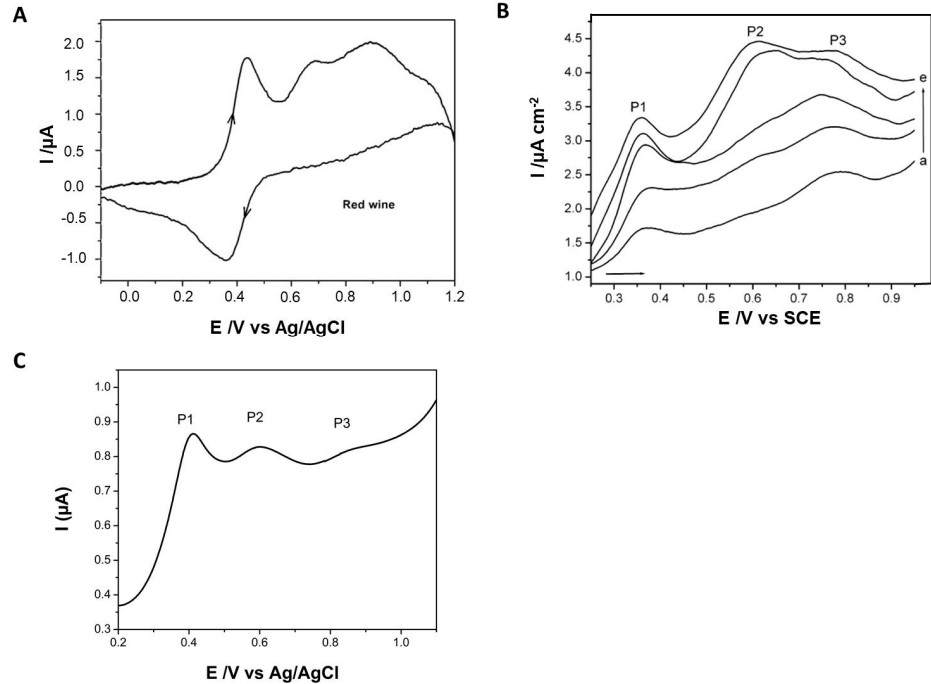

**Figure 3.** (**A**) Cyclic voltammogram recorded with at a 3 mm GC electrode at 100 mV s$^{-1}$, for a red wine diluted 100-fold with a model wine solution. (**B**) Differential pulse voltammograms of red wines: (a) Frankovka, (b) Pinot Noir, (c) Zweigelt, (d) Plavac Hvar, (e) Ivan Dolac, diluted 1/400 in acetate buffer solution pH 3.6, measured at the GC electrode; scan rate, 5 mV s$^{-1}$. (**C**) Square wave voltammogram of a red wine, diluted 60-fold in the model wine solution, pH 3.6 on GC electrode at frequency 25 Hz and with a step potential of 4 mV. Reproduced from [8] (**A**), and [11] (**B**), with permission from Elsevier.

The peak current density of P1 (Ip$_1$) and the charge passed up to 500 mV (Q500) were shown to be highly correlated with the total phenolic compounds (TPC) determined by the Folin–Ciocalteu method [4,11]. However, the absolute values recorded by electrochemical methods and the Folin–Ciocalteu were different, even when using the same standard compounds (e.g., gallic acid or catechin) for expressing the results. The differences between the absolute concentrations of TPC determined via electrochemical methods and by the Folin–Ciocalteu test are well-known [7,11,73] and are attributed at least in part to the reactions of other wine components, including sugars and sulfites with the colored Mo(VI)-based reagent in the Folin–Ciocalteu test. As long as the electrochemical method is robust and its correlation with Folin–Ciocalteu is established, the electrochemical sensors can be used for monitoring the progress of the maceration-fermentation of red wines, or to determine changes induced by ageing, different technologies and treatments along wine production.

New opportunities to determine quantitatively more composition parameters besides TPC and to discriminate among wines arise by modifying the electrode surface with mediators, nanomaterials, conducting polymers or molecularly imprinted polymers. Voltammograms acquired with sets of sensors modified with various mediators or made from different electrode materials provide a unique "fingerprint" of a wine sample. These sensors are assembled in e-tongue devices where the input data from the voltammetric sensors is interpreted via chemometric methods for either qualitative (discrimination) or quantitative purposes (determination of composition parameters). The group of Rodriguez-Mendez has worked extensively on such devices devoted to the wine industry [22].

While the voltamperometric methods based on the direct oxidation of phenolic compounds on carbon electrodes have numerous advantages compared to spectrophotometric tests related to the possibility to measure colored turbid samples, in order to assess the TPC content in a very fast manner requiring low-cost equipment, it should be noted from a practical point of view that phenols' oxidation

leads to electrochemically inactive compounds that are strongly adsorbed on the surface. Thus, the electrode surface has to be renewed after each test.

In this context, the use of nanomaterials brought many advantages in the determination of wine polyphenols and food antioxidants in general [57], including resistance to fouling, increased sensitivity, and selectivity. A representative example is the use of carbon nanotubes as surface modifier in carbon-based electrodes, enabling better electron-transfer kinetics for polyphenol oxidation and enhanced operational stability in FIA systems. The detection of wine phenolic compounds such as gallic acid, caffeic acid, ferulic acid and *p*-coumaric acid was achieved at lower overpotentials and with increased sensitivity on carbon nanotubes (CNT)-modified electrodes compared to bare GCE electrodes [3,65] (Figure 4A). In another study, a GCE modified with a suspension of MWCNT in polyethyleneimine (PEI) displayed good operational stability in FIA experiments, protection against fouling and against passivation due to the adsorbed oxidation product of polyphenols [3] (Figure 4B). Upon testing for 4 days at +0.7 V with 40 wine tests each day, the sensor response decreased to 70% of the initial value, which represents an appropriate stability for monitoring fermentation processes. The amperometric detection of either the fraction of easily oxidisable antioxidants in wine (mostly o-diphenols, determined at +0.3 V) or the total phenolic compounds (determined from the current intensity at +0.7 V) was straightforward by simply setting the potential at the appropriate value. The MWCNT/PEI/GCE sensor was characterized by detection limits of 0.07, 0.05, 0.08 and 0.09 $\mu$mol L$^{-1}$ for caffeic acid, gallic acid, ferulic acid and *p*-coumaric acid, respectively with linear ranges extending up to 100 $\mu$mol L$^{-1}$. The sensor did not respond to ethanol, organic acids or sugars and only a small (4.4%) increase in signal was observed when testing with 0.1g L$^{-1}$ sulfite at +0.70 V. The lack of interference due to sulphite was explained by the particularities of FIA detection, the oxidized polyphenols being washed away by the electrolyte flow and thus unable to interact with sulphite. With this experimental setup it was found that the TPC determined via amperometric detection at +0.7 V expressed as mg L$^{-1}$ gallic acid was well correlated with the results obtained via the Folin–Ciocalteu test ($R^2 = 0.981$), as well as with the polyphenol index $I_{280}$ ($R^2 = 0.988$) and with the color intensity of wines ($R^2 = 0.962$).

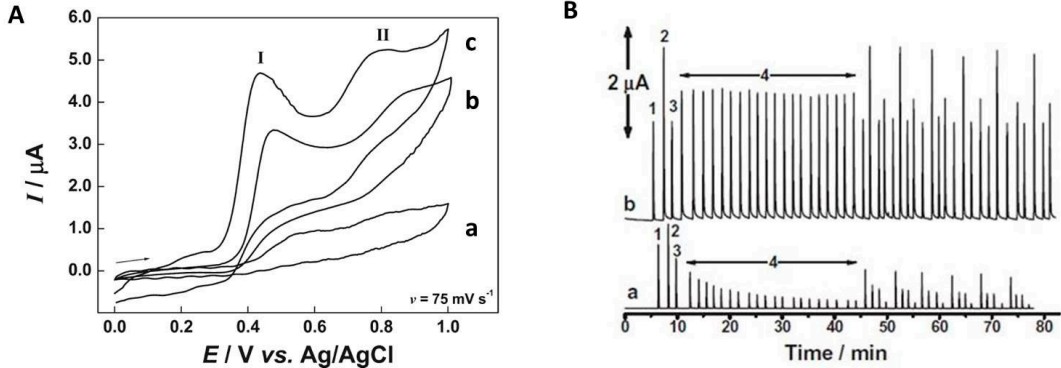

**Figure 4.** (**A**). Cyclic voltammograms for the electrochemical oxidation of $1.0 \times 10^{-4}$ mol L$^{-1}$ gallic acid in a pH 2.5 phosphate buffer solution at glassy carbon (a), carbon paste (b), and modified carbon paste (c). (**B**). Stability of the amperometric signal at +0.8 V of bare GCE (a) and GC/(CNT/PEI) (b) after successive injection cycles of 25 $\mu$M of caffeic (1), gallic (2), ferulic (3) and p-coumaric (4) acids. One cycle was followed by 20 successive injections of p-coumaric acid and then by six cycles of injections. Reproduced from [65], with permission from the American Chemical Society (**A**) and from [3], with permission from Elsevier (**B**).

The work on the amperometric detection of polyphenols with CNT-modified carbon electrodes in FIA systems [3], combined with the commercial availability of various screen-printed CNT-modified electrodes, paves the way for facile, cost-effective and robust analytical devices to be used in on-line systems for the monitoring of phenolic compounds and related parameters during wine production.

Besides providing increased stability and sensitivity, nanomaterials enabled enhanced selectivity compared to the unmodified electrodes as proven among others by the application of a nitrogen-doped carbon-modified glassy carbon electrode (NDC/GCE), for the determination of caffeic acid in wine [79]. The sensor displayed superior electrocatalytic activity featuring low resistance to charge transfer, sharp redox peaks and high anodic and cathodic peak currents for caffeic acid, observed at lower overpotential compared to GCE or graphene oxide-GCE. These superior characteristics were attributed to the electroactive pyridinic and pyrollic groups in the NDC lattice [79]. The operational and storage stability of the NDC/GCE electrode were very good: the anodic peak current intensity decreased to 93% of the initial value after 20 consecutive measurements by DPV and a similar decrease of only 6.5% from the initial value was observed after 6 weeks of storage The sensor was characterized by a detection limit of 2.4 nmol $L^{-1}$ and the repeatability of the measurements was very good, as indicated by RSD of 2.0–2.5% for n = 3 measurements of a wine sample. The claim of selective detection of caffeic acid is supported by voltammograms recorded for caffeic acid in mixture with possible interfering phenolic compounds (Figure 5A). While not discussed in detail, selectivity appears to be supported by higher detection sensitivity for caffeic acid and a slightly different oxidation potential at around 0.2 V compared to other phenolic compounds. Although other potential interfering compounds (e.g., catechol-CT, gallic acid-GA, ferulic acid-FA, ascorbic acid-AA and uric acid-UA) are oxidized in the same potential range (within 100 mV as apparent from the voltammogram in Figure 5A) and contributing to the shape of the voltammograms and a larger half-width of the anodic peak at around +0.2 V when in mixture with caffeic acid, the peak height, i.e., the peak current intensity remains largely unaffected. No interferences were found from 200 fold excess of $Na^+$, $K^+$, $Mg^{2+}$, $Ni^{2+}$, $Cu^{2+}$, $Cl^-$, $NO^{3-}$, $SO_4^{2-}$, glucose, dopamine and $H_2O_2$.

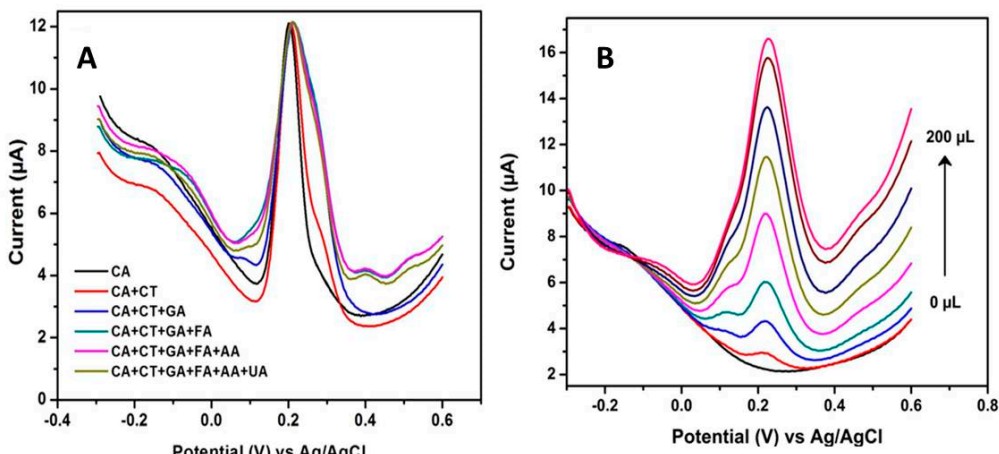

**Figure 5. (A)** The DPV responses of caffeic acid (CA) oxidation on nitrogen-doped carbon-modified glassy carbon electrode (NDC/GCE) in the presence of 10 times higher concentration of catechol (CT), gallic acid (GA), ferulic acid (FA), ascorbic acid (AA) and uric acid (UA). (**B**). The DPV responses for the determination CA in red wine sample with the NDC/GCE for increasing volumes of wine. Reproduced from [79] with permission from the authors, Creative Commons License 4.0.

For the analysis of wine samples with the NDC/GCE sensor, different volumes of wine were added incrementally in the electrochemical cell (Figure 5B) and the anodic peak current intensity was interpolated on the calibration curve for caffeic acid. However, the results obtained for five wines were not verified by a parallel analysis using a standard method to confirm the sensor accuracy.

In another report, electrode modification with the conducting polymer poly-3,4-ethylenedioxythiophene (PEDOT) led to enhanced sensitivity, antifouling capacity and selectivity, i.e., the electrochemical oxidation signals from ascorbic acid, phenolic compounds and sulphite can be resolved at PEDOT-covered electrodes [15,87].

Molecular imprinted polymers (MIP) provide yet another means for achieving selectivity in the determination of wine phenolic compounds. Leite et al. [77] described a sensor for caffeic acid made by modifying an Au electrode with molecularly imprinted siloxanes (MIS) obtained via a sol–gel process (Figure 6A).

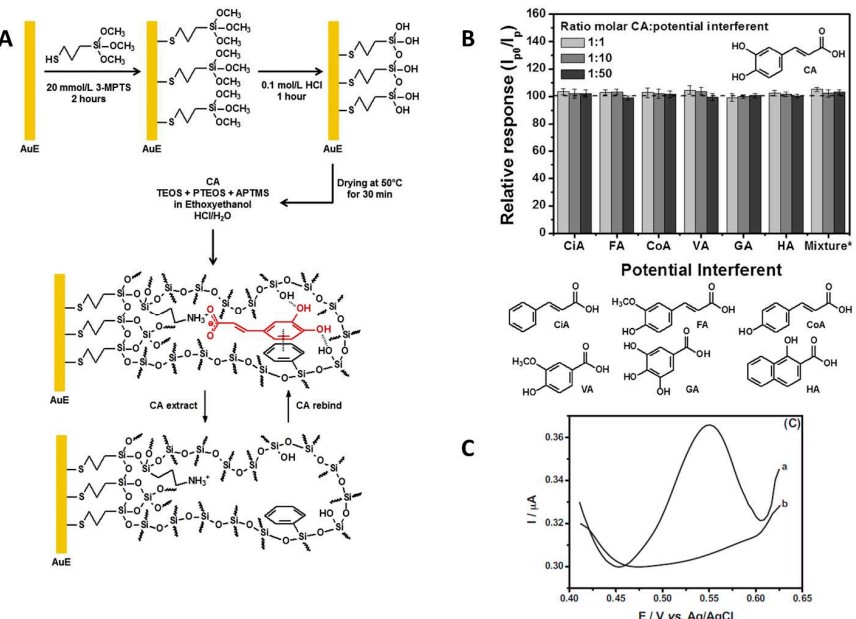

**Figure 6.** (**A**) Schematic illustration of the functionalization of the Au electrode with molecularly imprinted siloxanes. TEOS: tetraethoxysilane; PTEOS: phenyltriethoxysilane and 3-APTMS: 3-aminopropyltrimethoxysilane. (**B**) Interference study, showing the relative analytical response (Ip0/Ip) for 20 μmol L$^{-1}$ caffeic acid (CA) in presence of cinnamic acid (CiA), ferulic acid (FA), p-coumaric acid (CoA), vanilic acid (VA), gallic acid (GA), 1-hydroxy-2-naphthoic (HA). *Mixture of the five potential interferents at 20 μmol L$^{-1}$. (**C**) DPV responses for the MIS/AuE (curve a) and the non-imprinted siloxanes/AuE (curve b) after incubation in 50 μmol L$^{-1}$ CA for 30 min. Pulse amplitude: 25 mV, scan rate: 20 mV/s. Supporting electrolyte: 0.5 mol L$^{-1}$ H$_2$SO$_4$. Reproduced from [77] with permission from Elsevier.

After forming the MIS on the electrode surface in the presence of the template molecule, caffeic acid, the template was removed, leaving a cavity that specifically binds caffeic acid, with no interferences from cinnamic acid, ferulic acid, *p*-coumaric acid, vanilic acid, gallic acid, and 1-hydroxy-2-naphthoic acid (Figure 6B). The intensity of the anodic peak due to caffeic acid oxidation by DPV (Figure 6C) varied linearly with the concentration of caffeic acid in the range 0.500–60.0 μmol L$^{-1}$, with a detection limit of 0.15 μmol L$^{-1}$. Four wines were analyzed in parallel with the sensor and by a chromatographic method, the results being similar. The accuracy of caffeic acid detection in wines with the sensor was, moreover, confirmed by spiking experiments where the recovery factors for caffeic acid were 97.4%–102.3%.

Enzyme-Mediated Amperometric Detection

Various amperometric biosensors based on polyphenoloxidases (copper-containing oxidases such as tyrosinase and laccase) or horseradish peroxidase have been developed for the quantitative determination of phenolic compounds in wines.

Tyrosinase catalyses the oxidation of monophenols and o-diphenols to quinones in the presence of oxygen.

(1)　　Phenol + O$_2$ $\xrightarrow{\text{tyrosinase}}$ o-quinone + H$_2$O

Laccase catalyzes the oxidation of substituted mono- and polyphenols, aromatic amines and thiol compounds, leading to phenoxy radicals that can be further oxidized to quinones. In the same reaction, oxygen is reduced to water [88]:

(2)     $AH_2 + \frac{1}{2} O_2 \xrightarrow{\text{laccase}} A + H_2O$, where $AH_2$ and A represent the reduced and oxidized form of the polyphenolic compound, respectively.

Horseradish peroxidase (HRP) acts as a catalyst in the oxidation of phenols in the presence of hydrogen peroxide.

(3)     $Phenol_{red} + H_2O_2 \xrightarrow{\text{HRP}} Phenol_{ox} + 2 H_2O$

The phenoxyradicals and the quinones formed in the enzymatic reactions can be reduced electrochemically on the surface of a suitably polarized electrode and the magnitude of the reduction current is proportional to the amount of phenolic compounds in the sample. Alternatively, the oxygen or hydrogen peroxide consumed in the reaction can be determined electrochemically.

As summarized by Cortina-Puig et al. [58], laccase-based biosensors have a broader specificity and higher stability compared to devices relying on tyrosinase. Moreover, laccase biosensors are advantageous over HRP-modified electrodes which respond to a wider range of phenolic compounds but require the addition of $H_2O_2$ for the enzymatic reaction. Therefore, amperometric devices using laccase from either *Trametes, Aspergillus* and *Ganoderma* genera as a catalytic element were the preferred approach [89] for the determination of the total polyphenolic content of wines (Table 2).

The best-performing devices [89] feature covalently immobilized enzymes, thermostable enzymes and nanomaterials such as AuNPs, AgNPs and CNTs (improving the conductivity and providing a higher surface area and enzyme loading). Such characteristics ensure robustness, sensitivity and stability of the biosensors, providing high-throughput analysis when included in FIA systems, some biosensors being stable for up to 10 months and 900 measurements [89].

A 2006 report by Gamella et al. [72] described a biosensor made by immobilizing the enzyme from *Trametes versicolor* by cross-linking with glutaraldehyde on the surface of a glassy carbon electrode. While very simple, this biosensor displayed excellent analytical characteristics both in batch experiments and FIA setups. In FIA, for example, the linear ranges for gallic acid and caffeic acid were 0.04–2.0 mg $L^{-1}$ and 0.001–0.100 mg $L^{-1}$, respectively. The detection limits were 0.04 and 0.001 mg $L^{-1}$, respectively, and the biosensor had a good repeatability, proven by RSDs of 7.3 and 9.5% for 10 successive calibration plots for gallic acid and caffeic acid. The biosensor was not affected by glucose and ascorbic acid and the values of the polyphenol index measured for a series of calibration (standard) solutions of gallic acid and various white, rosé and red wines were highly correlated ($R^2$ = 0.997) with those measured in parallel via the Folin–Ciocalteu method. This amperometric biosensor, operating at -0.1 V at pH 5 allowed to observe a somewhat lower apparent Michaelis constant $K_m^{app}$ for caffeic acid than for gallic acid (8.2 ± 0.1 mg $L^{-1}$ and 9.8 ± 0.1 mg $L^{-1}$ for caffeic and gallic acid, respectively), indicating a higher affinity of the immobilized enzyme for caffeic acid. The sensitivity of the biosensor was much higher for caffeic acid compared to gallic acid (Figure 7). In a FIA setup, the sensitivity of the biosensor (the slope of the calibration curve) was (27 ± 2) × $10^{-2}$ µA mg$^{-1}$ L for caffeic acid versus (1.85 ± 0.06) × $10^{-2}$ µA mg$^{-1}$ L for gallic acid.

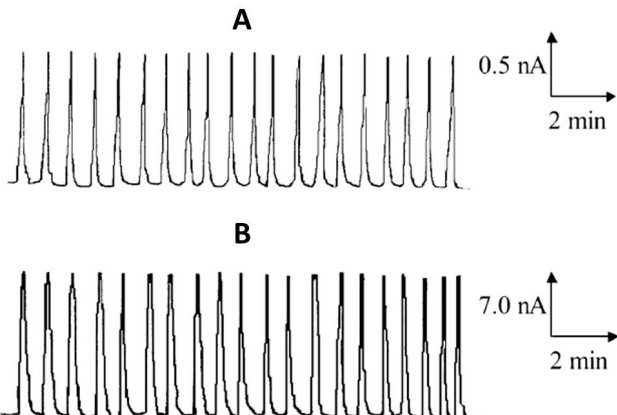

**Figure 7.** Flow injection amperometry (FIA) responses obtained at the laccase biosensor for 20 repetitive injections of 0.5 mg $L^{-1}$ gallic acid (**A**) and 0.05 mg $L^{-1}$ caffeic acid (**B**). Carrier solution was 0.1 mol $L^{-1}$ citrate buffer of pH 5.0. Flow rate: 0.3 mL $min^{-1}$, Vi: 150 μL, $E_{app}$: −0.20 V. Reproduced from [72] with permission from the American Chemical Society.

This is expected, considering the differences in the rate of electron transfer for the electrochemical reduction of quinone products formed at the electrode surface after the laccase-catalyzed oxidation of caffeic acid and gallic acid, respectively [6]. More recently, Vasilescu et al. [64] described the development of a biosensor made by modifying a carbon screen-printed electrode with laccase from *Trametes versicolor* and a nanocomposite formed by $MoS_2$ flakes and graphene quantum dots (GQDs). The nanocomposite provided a 2.2 times higher surface area compared to the bare electrode and facilitated the immobilization of negatively charged laccase by electrostatic interaction with the positively charged GQDs. The apparent Michaelis constants for chlorogenic acid, caffeic acid and catechin were 1.09 μmol $L^{-1}$, 11.25 μmol $L^{-1}$ and 121.80 μmol $L^{-1}$, respectively. The biosensor was applied for the analysis of total polyphenolic in red wines, the results being very similar to those obtained via the Folin–Ciocalteu method.

A stable biosensor, showing 87% of the initial activity after 120 tests in a FIA setup was obtained by modifying a gold electrode with layers of AuNPs, fullerenes and the same enzyme (laccase form *Trametes versicolor*) [63] (Figure 8).

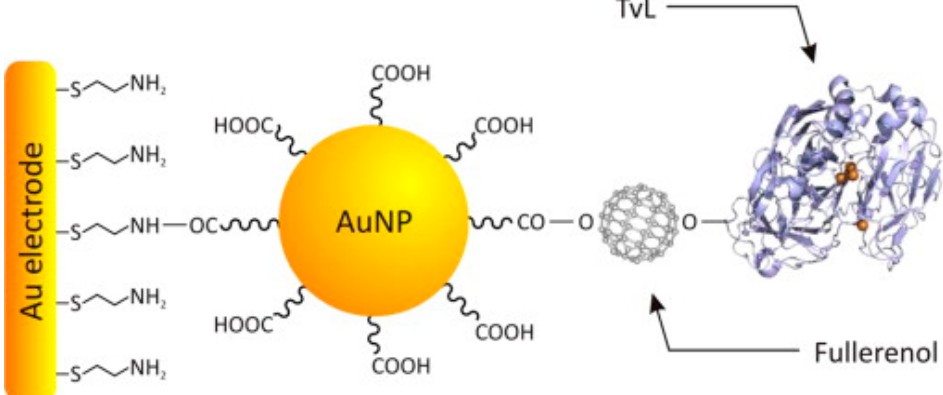

**Figure 8.** Gold self-assembled monolayer (Au-SAM)/gold nanoparticles (AuNPs)-Linker/Fullerenols/TvL composite material assembly. Reproduced from [63] with permission from Elsevier.

This work nicely illustrates the advantages of the nanomaterials used and their contribution to the sensor performances. A comparison of the electrochemical and kinetic parameters for biosensors obtained with (1) Au-SAM electrode, (2) Au-SAM/AuNPs and (3) Au-SAM/AuNPs/fullerenol shows that

the diffusionless electron transfer rate constant ($k_s$) and the amount of immobilized enzyme ($\Gamma'$) increases, in the order Au-SAM-laccase < Au-SAM/AuNPs-laccase < Au-SAM/AuNPs/fullerenol-laccase, while the $K_m{}^{app}$ for gallic acid decreases significantly in the same order ($k_s$ from 0.4 to 0.9 s$^{-1}$, $\Gamma$ from $6.6 \times 10^{-11}$ mol cm$^{-2}$ to $3.0 \times 10^{-10}$ mol cm$^{-2}$ and $K_m{}^{app}$ from 0.84 to 0.66 mmol L$^{-1}$). The AuNPs and the fullerenols increase the conductivity, surface roughness and provide a higher surface area for immobilization of the enzyme with preservation of its activity. Application of this biosensor for the analysis of the total polyphenol index in two wine samples (white and red) in 0.1 mol L$^{-1}$ acetate buffer, pH 4.5 with amperometric detection at −0.1 V has allowed calculating values of the total polyphenol index very close to those determined by the Folin–Ciocalteu method.

In one report on tyrosinase based biosensors for wine analysis, the enzyme was immobilized on the surface of sonogel carbon electrodes concomitantly with the electropolymerisation of EDOT (3,4-ethylenedioxythiophene), using a sinusoidal current method [90]. The PEDOT/tyrosinase/sonogel carbon biosensor had a $Km^{app}$ of 178.72 µmol L$^{-1}$, showing a high affinity for caffeic acid. A set of 4 wine and 9 beer samples were analysed by chronomaperometry at +0.17 V using the standard additions method and in parallel via the TEAC (ABTS (Total equivalent antioxidant capacity assay based on 2,2′ -Azino-bis(3-ethylbenzothiazoline-6-sulphonic acid) spectrophotometric method. For two red wines of the same brand but with different aging time in oak barrels (>24 months and >36 months, respectively) the polyphenol index measured with the biosensor was higher for the wine with a longer aging time (1186 µmol L$^{-1}$ versus 936 µmol L$^{-1}$ expressed as caffeic acid). The authors attributed this to the polyphenols extracted from the oak barrels [90]. The estimated cost for the biosensor was under 2 euros and the fabrication time was under 5 min. No decrease in biosensor performances was observed after storage for 10 days at 4 °C [90].

Considering the different sensitivity and selectivity of the biosensors based on different enzymes, Cetó et al. [68] proposed a voltammetric (bio-)electronic tongue for the determination of total phenolic compounds and discrimination of individual phenolic compounds in wines. The device included 2 chemical sensors (graphite-epoxy and Cu NP-modified) and 2 biosensors (based on tyrosinase and laccase). The cyclic voltammetry data obtained with the sensors for the analysis of 29 wine samples from Spain were used as input to build an artificial neural network (ANN) model. Predictions of the TPC based on this ANN model were well correlated with the results obtained in parallel by Folin–Ciocalteu and ultraviolet (UV) polyphenol index ($I_{280}$). Remarkably, chemometric analysis of wines spiked with various individual phenolic compounds, coupled with chemometric analysis by principal components analysis-ANN allowed the different polyphenols to be distinguished.

Nanomaterial–Enabled Biomimetic Detection of Polyphenolic Compounds

The detection of wine polyphenols, mainly those with o-diphenol groups, was achieved with ceria nanoparticles showing oxidase-like activity and acting as mimetics for laccase and tyrosinase. Ceria nanoparticles, with dual 3+ and 4+ oxidation states and surface oxygen vacancies can oxidize compounds with o-dihydroxybenzene groups to the corresponding quinones, moreover these quinones can form charge transfer complexes at the surface of nanoceria increasing their local concentration. Such mechanism, supported by Fourier transform infrared (FTIR) and CV data, were indicated for a sensor where ceria NPs were dropcasted at the surface of screen-printed carbon electrodes (SPCE) [67]. The quinones formed from the interaction of polyphenols with nanoceria are reduced amperometrically at a low applied potential of −0.1 V, where interferences due to the direct electrochemical transformation of wine components are minimal (Figure 9). The single use CeNP/SPCE sensor responded to a series of antioxidants found in wines (caffeic acid, gallic acid, quercetin, ascorbic acid) in the µmol L$^{-1}$ range, being similar in terms of detection limit and linear range to oxidase-based biosensors for the same phenolic compounds [67].

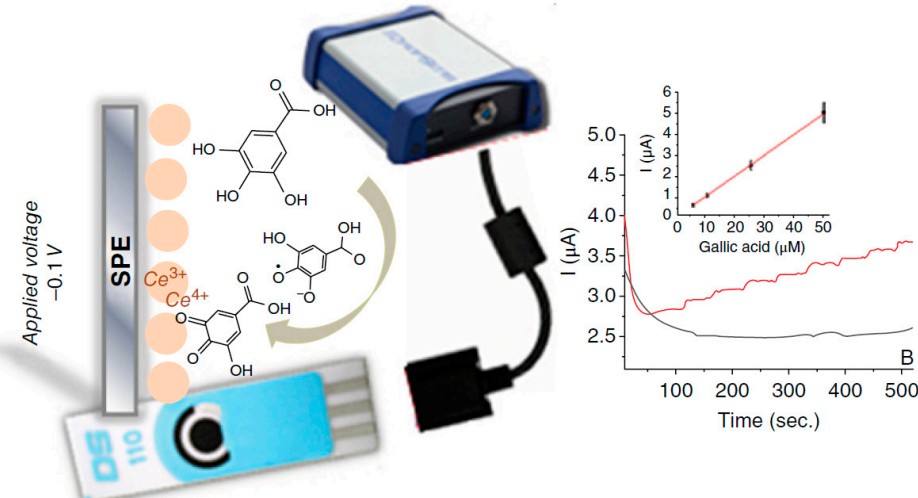

**Figure 9.** Principle of the nanoceria modified sensor for the determination of antioxidants. Reprinted from [67] with permission from Elsevier.

The sensor allowed differences between Romanian red wines of the same variety and harvest that were produced by different maceration-fermentation technologies to be observed.

More recently, Tortolini et al. have compared the electrochemical properties and sensitivity to gallic acid for 3 types of screen-printed electrodes (SPE) modified with ceria nanoparticles (CeNP) by drop-casting [91], namely (i) electrodes with nanostructured carbon, (ii) with carboxylic acid functionalized multi-walled carbon nanotubes (MWCNT) and (iii) with carboxylic acid functionalized multi-walled carbon nanotubes-$Fe_3O_4$ superparamagnetic nanoparticles. The authors found that the voltammetric sensor with carboxylic acid-functionalized multi-walled carbon nanotubes (MWCNT/CeNP/SPE electrode) provides the highest electrochemical area and best sensitivity to gallic acid. The sensor was characterized by detection limits of 7 $\mu$mol L$^{-1}$, 10 $\mu$mol L$^{-1}$, 9 $\mu$mol L$^{-1}$, 8 $\mu$mol L$^{-1}$ and 7 $\mu$mol L$^{-1}$ for the determination of gallic acid, caffeic acid, quercetin, trans-resveratrol and ascorbic acid, respectively by SWV. The analysis of 6 six wines with the MWCNT/CeNP/SPE electrode gave similar absolute values of the TPC expressed as gallic acid with the total antioxidant capacity (TAC) determined by the TEAC method and expressed in the same way, despite the different mechanisms behind the two methods.

These two examples emphasize the role of electrodes' surface functionalization and experimental conditions such as pH, detection method and choice of standard antioxidant compound used as reference in the detection of wine polyphenols. In addition, the oxidase mimetic properties of nanoceria depend on NP size and zeta potential [92]. Definitely, the potential of electrochemical sensors based on ceria NPs for determining the polyphenol content of wines is only starting to be uncovered.

### 2.3.2. Biosensors for the Total Antioxidant Capacity (TAC)

According to a widely accepted definition an antioxidant is "any substance that delays, prevents or removes oxidative damage to a target molecule" [93]. The antioxidant effect can be achieved in different ways, such as through the scavenging of free radicals and molecular oxygen, by chelating metals such as Cu and Fe or inhibiting the activity of enzymes involved in the production of reactive oxygen species (xanthine oxidase, lipoxygenase). The phenolic compounds in grapes and wine are powerful antioxidants thanks to their structure and their ability to scavenge free radicals, chelate metals etc. and the tests by which these abilities can be tested were amply investigated [20,58,60,94–96]. Since the antioxidant capacity is produced by multiple mechanisms, there is not a single method that best quantifies the total antioxidant capacity of wines (or another matrix, for that matter). The most frequently used methods for the detection of antioxidant capacity are ORAC (Oxygen radical absorbance capacity),

FRAP (Ferric reducing ability of plasma), TRAP (Total Reactive Antioxidant Potential), CUPRAC (Cupric ion reducing antioxidant capacity), ABTS•[+] and DPPH•(2,2-Diphenyl-1-picrylhydrazyl assay), based on optical detection by fluorescence, chemiluminescence or colorimetry [57] along with a series of electrochemical methods [4,20,57,60,97].

The electrochemical determination of the TAC of wines encompasses several approaches: (1) calculation of different electrochemical index indicators from the current intensity-potential curves recorded for the direct oxidation of wines on carbon electrodes, (2) adaptation of classic spectrophotometric methods such as those based on ABTS•[+] and DPPH• for electrochemical detection, (3) biosensors based on enzymes such as superoxide dismutase or cytochrome.

As in the case of total phenolic compounds, the voltamperometric (bio)sensors for the determination of TAC were typically applied to a limited set of wine samples being manly focused on the correlation of the newly proposed procedures with more established methods:

(1) calculation of electrochemical index indicators from the intensity-potential curves for the direct oxidation of wines on carbon electrodes.

According to their structure, polyphenols can be electrochemically oxidized at different potentials. Those oxidized at low overpotentials have high electron-donating capacity, hence higher antioxidant capacity. Indeed, a good correlation was found between the current intensity at the potential corresponding to the first oxidation peak (P1) of wines on carbon and CNT-modified modified electrodes and the TAC determined via various spectrophotometric methods [65]. Some authors have proposed electrochemical indexes that would reflect more closely the variability of wine polyphenolics including both compounds with high electron-donating capacity (oxidized at low overpotential) and lower electron ability (oxidized at high overpotential). For example, Lino et al. [98] considered also the correlation of the intensity of anodic peak current with the amount of polyphenols oxidizing in that potential range and have proposed an electrochemical index EI as a measure of wine TAC, defined as:

$$EI = \frac{Ipa1}{Epa1} + \frac{Ipa2}{Epa2} + \frac{Ipa3}{Epa3}$$

where Ipa and Epa represent the anodic peak current intensity and the anodic peak potential, respectively for the 3 major peaks found in the voltammograms of red wines.

Based on extensive analysis by DPV on carbon paste electrodes of red, rose and white wines, a fair correlation ($R^2$ = 0.9110) was found between the EI index and EC50, representing the amount of wine needed to produce a 50% discoloration of a DPPH• solution compared to the control.

(2) adaptation of classic spectrophotometric methods such as those based on ABTS•[+] and DPPH• for electrochemical detection, taking advantage of the electroactivity of these radicals.

Wine polyphenols and antioxidants in general have different kinetics for their reaction with these synthetic radicals [99]. As a result, the sample dilution and the incubation time influence the results of the TAC assays, in addition to other factors such as the applied potential, the pH, and the electrolyte. For example, trolox, used widely as a standard compound for expressing the TAC, has a fast scavenging of DPPH• that is not influenced by pH in the range 4–7. In contrast, wine polyphenols interact more slowly with DPPH• and their radical scavenging activity is pH-dependent, with higher TAC determined in acidic media. Due to the different experimental conditions used by different authors, it is difficult to compare the different assays and antioxidants. One solution proposed by Magalhaes et al. [100] for a more universal and relevant indicator of TAC is the "kinetic matching approach" where a mixture of phenolic compounds relevant for the specific food matrix (e.g., caffeic acid, catechin, hesperetin, morin and (-)- epigallocatechin gallate) is used for the calibration, rather than a single compound, and the reaction with the sample is allowed to proceed for 10 min.

With regard to applications in monitoring wine production, Andrei et al. [69] have described the amperometric detection of DPPH•·for the determination of TAC in wines using a FIA system and a screen-printed 3 electrode system including a gold working electrode. Based on a 3 min incubation of wine samples with a 100 μmol L[−1] DPPH• solution and detection at −0.1 V in a 1:1

mixture of phosphate buffer pH 6 and ethanol, the sensor responded linearly in the range 2–30 µmol $L^{-1}$ trolox, with a detection limit of 0.45 µmol $L^{-1}$. The test was shown to be equivalent to the spectrophotometric method based on the same radical (DPPH•) and was highly correlated with both the trolox equivalent antioxidant capacity test based on ABTS•+ radical ($R^2$ = 0.966) and with the TPC assay by Folin–Ciocalteu ($R^2$ = 0.93). Moreover, the TAC was correlated fairly well ($R^2$ = 0.88) with the amount of tannins, but not with the concentration of anthocyanins in wines. The assay was used to evaluate the changes in the antioxidant capacity of wine at different technological steps (i.e., after the malolactic fermentation, at 1 week and at 3 months after bottling), for two types of red wines obtained by different technologies: classical (1), with partial running off of the juice (2), maceration with daily remontage (3); maceration-fermentation with addition of tannin (4), delestage (5) and fermentation in the presence of pectolytic enzymes (6). All wines had high TAC (5–8 mmol $L^{-1}$ trolox) and for 11 of the 12 samples tested the TAC decreased 3 months after bottling as compared to the initial value. The results emphasized differences in the TAC of wines from the 2 grape varieties but no significant differences due to the maceration-fermentation techniques used.

(3) biosensors based on enzymes.

The measurement of a phenolic compound's ability to scavenge the superoxide radical formed in situ in the reaction medium is considered to reflect more closely its potential effects in the human body than methods based on artificial radicals. While a whole body of work was devoted to electrochemical biosensors for measuring the burst of free radicals (reactive oxygen species, ROS) in biological media, their application for the evaluation of the antioxidant activity of wines was limited [101]. This is mainly due to the complexity, cost and insufficient stability of the biosensors combined with the abundance of alternative chemical sensors. Campanella et al. [101] developed a superoxide dismutase (SOD) biosensor to evaluate the antioxidant capacity of wines. The measurement was based on the capacity of wine antioxidants to scavenge the superoxide radical produced in situ by adding xanthine and xanthine oxidase as per the reaction:

$$\text{xanthine} + H_2O + O_2 \xrightarrow{\text{xanthine oxidase}} \text{uric acid} + 2H^+ + O^{2-}$$

The superoxide radical is transformed in oxygen and hydrogen peroxide by the SOD immobilized at the biosensor surface.

$$O^{2-} + O^{2-} + 2H^+ + \xrightarrow{\text{superoxide dismutase}} H_2O_2 + O_2$$

Finally, the $H_2O_2$ formed was determined amperometrically, by polarizing the biosensor at a suitable potential for the electrochemical oxidation of $H_2O_2$. The intensity of the current generated in the presence of wine phenolic compounds is lower, since the antioxidants scavenge the superoxide radical, decreasing its concentration. The "relative antioxidant capacity" (RAC) of wines was defined by comparing the slope of sensor response to xanthine additions in the range 0.02–2 mmol $L^{-1}$ in the presence or absence of wine. The SOD biosensor was applied for testing 3 white and 4 red wines and the results were compared with spectrophotometric and spectrofluorimetric (ORAC) methods for TAC. While all methods allowed observing the same "ranking" of wine samples based on their TAC, the magnitude of the absolute differences in TAC between white and red wines depended on the measurement method. The authors have, moreover, developed a tyrosinase biosensor, a sulfite oxidase biosensor and an ascorbate oxidase biosensor to determine the total polyphenol, sulfite and ascorbic acid content in wines, as main contributors to wine TAC. Next, by measuring with the SOD biosensor the antioxidant capacity due exclusively to ascorbic acid and sulfite, the authors proposed to consider the "net TAC" of wines, where the contribution of sulfite was subtracted from the value measured with the SOD biosensor. Although the presumption of an additive contribution to TAC might be an oversimplification and the work focused on a limited set of only 7 wines, the work of Campanella et al. [101] nicely illustrates the relative contributions due to phenolic compounds, sulfite and ascorbic acid to the TAC of wines (Figure 10).

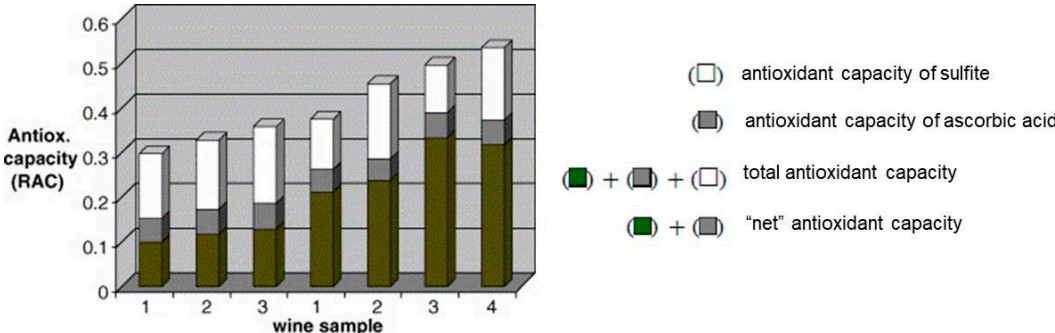

**Figure 10.** "Net" antioxidant capacity of wine samples (in relative antioxidant capacity, RAC, units), emphasizing the relative contributions of sulfite and ascorbic acid to the total antioxidant capacity of wines measured with the superoxide dismutase biosensor. Reproduced from [101] with permission from Elsevier.

## 2.4. Biosensors for Allergens

The group of additives used in the wine production that pose an allergen risk includes proteins from milk, eggs, fish or cereal and sulfur dioxide, used as a preservative and antioxidant. The applications of allergenic proteins in enology were reviewed by Penas et al. [102]. In addition to casein, gelatin, ovalbumin used exclusively for wine fining, lysozyme, a protein from egg white is approved for use as an antimicrobial agent, partially replacing sulfur dioxide. The maximum dose of lysozyme recommended for wine treatment is 500 mg L$^{-1}$, depending on the purpose (e.g., the control of the malolactic fermentation or the microbiological stabilization of wine) [103]. To protect the health of allergy-susceptible people, a suitable allergen warning has to be present on the wine label, according to the regulations in the European Union (EU) and worldwide. For egg proteins, their presence in wine has to be indicated if detected in the final wine by a method characterized by a detection limit of 0.25 mg L$^{-1}$ and a quantification limit of 0.5 mg L$^{-1}$, which represent analytical performances set by the OIV [104]. Therefore, it is important to develop and validate sensitive analytical methods meeting these performances. The standard methods for the detection of proteins in wine are based mostly on enzyme-linked immunosorbent assay (ELISA), chromatography, capillary electrophoresis, ELISA or polymerase chain reaction (PCR), various detection kits being available commercially [53,102,105]. The progress in allergen detection and the trend towards consumer-friendly technologies are remarkable, including personal use devices to test the presence of allergen in food (Sensogenic for eggs and peanuts from SensoGenic, Israel, the Nima sensor for gluten and peanuts from Nima, USA etc.). Between the devices for personal use and the regulated quality control environment specific to industrial processes, biosensors were proposed as fast and more cost-effective alternatives compared to the ELISA and PCR methods used for the quantitative assessment of the allergen risk [106].

With regard to the proteins used as processing aids in wine production, several electrochemical biosensors as well as bioassays coupled with electrochemical detection were proposed for the detection of milk and egg proteins (casein, B-lactoglobulin, ovalbumin, lysozyme) [107]. However, only a few of these sensors were tested so far with wines: e.g., optical biosensors for lysozyme [108–110] and ovalbumin [111] and electrochemical ones for lysozyme [10,110,112]. The development of biosensors for lysozyme was prompted by the availability of several aptamer sequences specific for lysozyme which can bind lysozyme with high affinity (affinity constants in the nM range). Aptamers, short oligonucleotide or peptide sequences selected in vitro to bind with high affinity and specificity to a selected target, are more stable compared to antibodies and their production is more cost-effective and reproducible. Since the first report on a selected lysozyme aptamer in 2001, lysozyme was a preferred model analyte when developing novel biosensing strategies involving different types of detection formats (e.g., direct, competitive, sandwich). Among the various electrochemical biosensors [113],

several were applied for wine analysis and used voltamperometric detection [10,112], as illustrated in Figure 11.

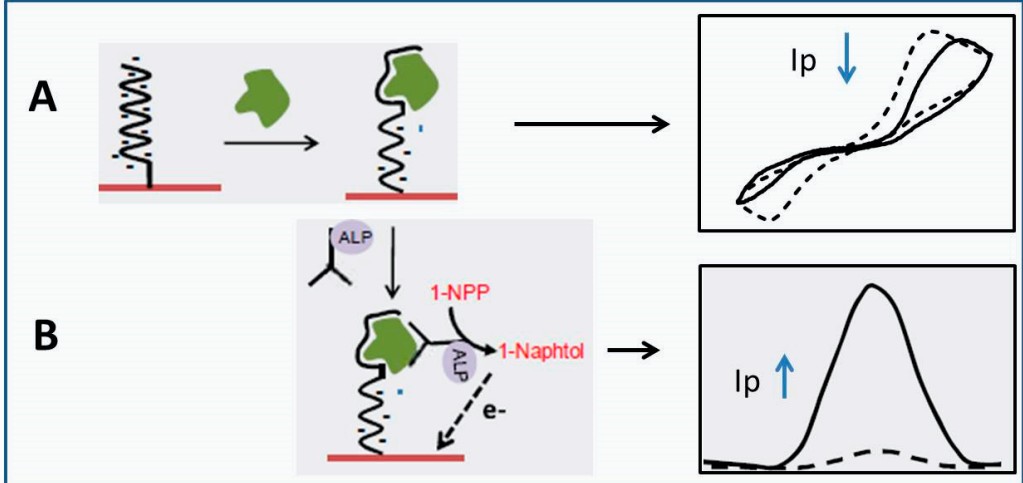

**Figure 11.** (**A**) Direct analysis of lysozyme: the formation of the affinity complex between the aptamer immobilized on electrode surface and lysozyme creates a barrier for the diffusion of a redox reporter (e.g., ferrocyanide) towards electrode surface. Hence, the anodic peak current due to the oxidation of ferrocyanide decreases, proportionally with the quantity of lysozyme in the sample. (**B**) Sandwich assay, where lysozyme binds both to the capture aptamer immobilized on electrode surface and to a detection antibody labeled with alkaline phosphatase (ALP). Lysozyme binding to the sensor leads further to the binding of the antibody, thus to more enzyme at electrode surface able to catalyze the transformation of 1-naphtyl phosphate (NPP) into 1-naphtol. The increase in the intensity of the current due to the electrochemical oxidation of 1-naphtol ($I_p$) is directly correlated with the concentration of lysozyme in the sample.

Ocana et al. developed a biosensor that combined a capture aptamer immobilized on the electrode surface with a detection antibody, in a sandwich detection format [112]. The antibody was labeled with the enzyme alkaline phosphatase (ALP) and the detection relied on the transformation of 1-naphtol, catalyzed by ALP and monitored by DPV. The detection limit was excellent (4.3 fM). However, for practical purposes it should be noted that this test has a certain level of complexity, presumes several steps and the use of biological elements (antibody and enzyme) of limited stability, in addition to the aptamer. Our group has developed recently a simple electrochemical aptasensor where the aptamer was immobilized by chemisorption on a screen-printed gold electrode modified with AuNP [10]. The detection limit achieved with the aptasensor was 0.32 µg mL$^{-1}$ (22 nM) lysozyme and the linear range was 1–10 µg mL$^{-1}$. Most importantly, the sensor was applied to monitor the variation of lysozyme concentration in wine during critical stages in winemaking for five experimental wines made by partially replacing sulfur dioxide with different quantities of lysozyme (Figure 12). The results were in agreement with data measured via a standard chromatographic method, thus proving the accuracy of the sensor.

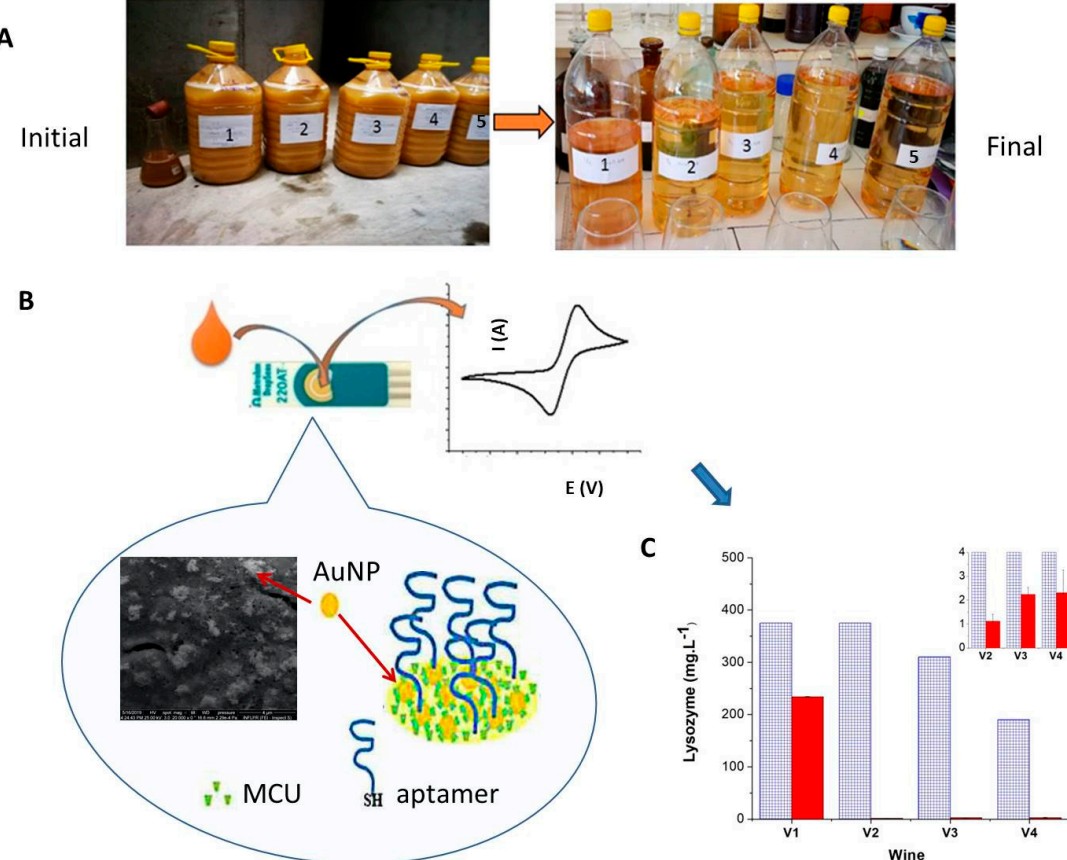

**Figure 12.** Application of an electrochemical aptasensor in the study of lysozyme–treated wines. (**A**) Experimental variants of must treated with different concentrations of lysozyme and the final wines obtained. (**B**) Design and principle of the aptasensor test: screen-printed electrode modified with AuNP and aptamer. The decrease in the anodic peak current due to ferrocyanide is proportional with lysozyme concentration in the sample. (**C**) Variation of lysozyme levels in four experimental wines V1–V4, measured with the aptasensor. Blue: initial, red: after the final conditioning step. Inset: zoom-in for V2–V4.

The aptasensor is simple and robust and has appropriate analytical performance for application in monitoring the wine production. While its sensitivity is not great compared to other biosensors for lysozyme, the surface modification with gold nanoparticles (AuNP) was an important feature for ensuring good analytical performance through an increased conductivity and electrochemically active area and a higher loading with aptamers. The electrochemical area increased from $0.058 \pm 0.006$ cm$^2$ to $0.115 \pm 0.010$ cm$^2$ and the average aptamer packing density increased from $1.67 \times 10^{12}$ to $3.52 \times 10^{12}$ molecules cm$^{-2}$ after the in situ electrochemical generation and deposition of AuNP.

Moreover, the study proved that the aptasensor is a useful tool to evaluate the allergen risk in relation to different processes in wine production. For example, it enabled in wines produced by adding lysozyme at the grape-crushing stage measurement of the residual amounts of lysozyme after the conditioning step by bentonite fining and filtration, which were around 1–2 µg mL$^{-1}$ (wines V2–V4, see inset in Figure 12C). In contrast, in variant V2 where lysozyme was added after the alcoholic fermentation, although lysozyme concentration decreased significantly after the condition treatment with bentonite, the residual amount was important, 234 µg mL$^{-1}$ and the allergen risk was considerable (Figure 12C). The study included a detailed physico-chemical and organoleptic characterization of wines obtained by various treatments with lysozyme, which facilitated the correlation between the characteristics of the wines obtained with lysozyme treatment [10].

In addition to egg and milk proteins which are not very often used as processing aids, the most prevalent allergen in wines is sulfur dioxide, added in both red and white wines for microbiological stability and to prevent oxidation (particularly in white wines). Regulatory agencies recommend including on the wine label the allergen warning "contains sulphites" for all wines with more than 10 mg L$^{-1}$ sulfite, expressed as SO$_2$.

For enologists, the determination of both total and free SO$_2$ is important for predicting the stability of wine and for establishing appropriate corrections. Unbound SO$_2$ in wine reduces the quinones formed by the electrochemical oxidation of catechol-containing phenolic compounds, a process linked to the main oxidation peak at around 400 mV in wines observed by CV with different, mainly carbon-based electrodes [8,114]. Thus, the catechol groups are regenerated and become able to undergo electrochemical oxidation again. Consequently, the intensity of the anodic peak at 400 mV is higher in the presence of free SO$_2$ [86]. The voltammetric detection of free SO$_2$ in white wines was achieved by taking advantage of the fast binding reaction between acetaldehyde and free SO$_2$, by comparing the voltammograms before and after the addition of acetaldehyde recorded on either bare GC electrodes or on PEDOT-modified Au [8,82].

The voltammetric detection of sulfur dioxide at unmodified carbon-based electrodes suffers, however, from strong interferences due to phenolic compounds in wine which are oxidized in the same potential range [4]. One possibility to determine the sulfites selectively is by using silver inkjet-printed electrodes modified with gold nanoparticles [9].

An alternative approach relies on cathodic stripping voltammetry at hanging mercury drop electrode (HMDE) to determine selectively not only sulfite, but also ethanethiol and inorganic sulfide [115]. The sample pre-treatment consisted in a 24 h stirring with a strong cation exchange resin with the purpose of eliminating the interference in the voltammetric measurement due to heavy metals. During the accumulation step, the oxidation of mercury in the presence of sulfite in wine leads to the formation of soluble Hg(SO$_3$)$_2^{2-}$ or insoluble Hg$_2$(SO$_3$) complexes which are reduced back to mercury during the stripping step. As shown in Figure 13 [115], ethanethiol, inorganic sulfide, and sulfite can be detected with a good resolution by this approach. The highest levels of ethanethiol, inorganic sulfide, and sulfite found in the Brazilian wines measured in this study were 4.85, 0.44, and 44.72 mgL$^{-1}$, respectively.

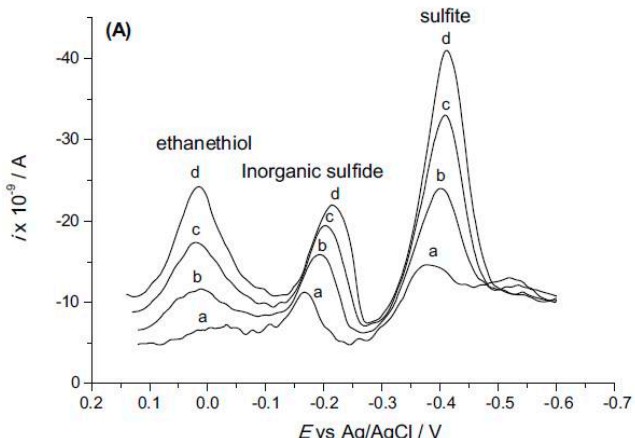

**Figure 13.** Voltammograms of Cabernet Sauvignon 2012. Experimental conditions: E$_{pt}$ = 120 mV, t$_{pt}$ = 30 s, E$_{start}$ = 150 mV, E$_{end}$ = −600 mV, $\upsilon$ = 10 mV s$^{-1}$. Standard concentration: ethanethiol: a sample, b 0.2, c 0.4, and d 0.6 mg L$^{-1}$; inorganic sulfide: a sample, b 0.016, c 0.032, and d 0.048 mg L$^{-1}$; sulfite: a sample, b 1.5, c 3, and d 4.5 mg L$^{-1}$. Reprinted from [115] with permission from Springer Science+Business Media.

Yet another possibility is to use biosensors based on sulfite oxidase (SOx) immobilized on the surface of screen-printed carbon electrodes for the selective detection of sulfite [116,117] in wines. The enzyme

catalyzes the oxidation of sulfite to sulfate in the presence of oxygen, with production of $H_2O_2$. The $H_2O_2$ can be determined electrochemically, using different mediators such as tetrathiafulvalene [116] to decrease overpotential and avoid interfering with compounds. Alternatively, the SOx was immobilized on an electrode surface together with a Os redox polymer by cross-linking in a mixture of bovine serum albumin (BSA) and cross-linking agent poly-ethylene glycol-400-diglycidyl ether (PEGDGE). The Os redox polymer functions as a mediator and a "wire", electrically connecting the enzyme with the electrode surface. Upon converting sulfite to sulfate, the enzyme is reduced. The reduced enzyme is then re-oxidised by the Os mediator, which becomes reduced from the Os (III) to the Os (II) form. In the final step, the mediator is oxidized electrochemically at −0.1 V and the current intensity is proportional to the amount of sulfite in the sample [117]. The biosensor kept 99% ± 5% of the initial performance after 4 days of operation in a FIA system, responded linearly to sulfite in the range 1–100 μm, had a response time of 2 min and a good repeatability, as indicated by an RSD of 4.8% for n = 12 measurements. The recovery of sulfite from 10 wine samples (white and red) spiked with 10 μM sulfite ranged between 63 ± 3 and 103 ± 6%. While further optimization and verification of performance by parallel testing via a standard method appear necessary, the biosensor is a promising tool for a high-throughput determination of sulfite. The drawbacks related to enzyme and mediator costs and lack of commercial availability, combined with the wide application of simple, titration-based methods and the existence of alternative electrochemical sensors as described above are limiting factors for a wider development and application of SOx biosensors in wine analysis.

## 2.5. Commercial Electrochemical Biosensors for Wine Monitoring

Most process control sensors for wine production available commercially address the alcoholic fermentation, where besides temperature, the main parameter to be monitored is the concentrations of sugars. This is mainly achieved by measuring the density of must and wine, with sensors installed outside the fermentation tanks and based on different principles, e.g., VS3000, Li Liquiphant M vibrating fork, Fermetrol probe, Micro-LDS (Integrated Sensing Systems Inc., USA), etc. [118]. Mid- and near-infrared (IR) sensors (e.g., commercial ones such as VS-3000 or the Alcolyzer Wine M/ME-Wine Analysis System produced by Anton Paar (Graz, Austria, www.anton-paar.com)) can be used to measure alcohol, $CO_2$ and specific gravity. Various flowmeters and differential pressure sensors used in-line to monitor the alcoholic fermentation are available from Endress Hauser (Reinach, Switzerland, www.endress.com). In addition, Raman spectroscopy can be used to measure glucose and ethanol during the fermentation processes [119]. Besides these sensors covering the basics for controlling the alcoholic fermentation at winery level, the detailed monitoring of wine production requires knowledge of many composition parameters. For this purpose, winemakers can currently rely on several analyzers based on electrochemical biosensors (Table 3). Analyzers with optical detection, as well as application-specific ELISA and immunochromatographic kits for oenology applications are also available from different vendors such as Biosentec (Auzeville-Tolosane, France, www.biosentec.fr), OptiEnz Sensors, LLC (Fort Collins, Colorado, USA, www.optienz.com), Biosystems S.A (Barcelona, Spain, www.biosystems.es), Megazyme (Bray, Ireland, www.megazyme.com), Accuvin LLC (Napa, California, www.accuvin.com/), R-Biopharm (Darmstadt, Germany, www.r-biopharm.com), Sigma-Aldrich (now Merck, Darmstadt, Germany, www.sigmaaldrich.com) etc.

**Table 3.** Commercial analyzers, biosensors and enzymatic, enzyme-linked immunosorbent assay (ELISA) and immunochromatographic kits for oenology.

| Biosensor Based System | Manufacturer | Detection Principle | Parameters | Type of Equipment |
|---|---|---|---|---|
| Biowine[300], Biowine[500], Biowine[700], | Biolan (Bizkaia, Spain, www. biolanmb.com) | Amperometry | Gluconic acid, malic acid, lactic acid, sugars, sucrose, histamine | Single to repeated use; portable + bench top |
| YSI 2900 Series Biochemistry Analyzer | Yellow Spring Instruments (Yellow Springs, Ohio, USA, https://www.ysi.com) | Amperometry, platinum electrode; membrane with immobilized enzyme | Glucose, Lactate, Glutamate, Glutamine, Glycerol, Xylose, Choline, Hydrogen Peroxide, Sucrose, Ethanol, Methanol, Lactose, Galactose | Bench top |
| OLGA-The On-Line General Analyser | Sensolytics GmbH (Bochum, Germany, www.sensolytics. com) | Amperometry | Glucose, Lactate Sucrose, Ethanol Glutamate | Sequential Injection Analysis (SIA)-system |
| LM5 lactate analyser; GL6 | Analox Instruments Ltd. (Stourbridge, UK, www.analox.com) | Amperometry, Clark-type oxygen electrode | Lactate, Ethanol, Glucose, Glycerol, Lactate, Methanol, Sucrose or Lactose. | Bench top |
| Handi-Lab biosensor measurement system | Gwent Group Advanced Materials systems (Pontypool, UK, www.gwent.org) | Amperometry | Glucose, Fructose | Single use sensors Portable |
| AMP Biosens | Biosensor SRL (Formello, Italy, www.biosensor-srl. eu) | Amperometry | Phenols, glucose, antioxidants | Bench top |
| Senzytec 2 | Tectronik srl (Limena, Italy, www.tectronik.it) | Amperometry | Ethanol, Malic acid D-Lactate, L-Lactate Glucose, Fructose | Portable |
| e-BQC | Bioquochem (Asturias, Spain, www.bioquochem. com) | Electrochemical | Antioxidant capacity | Portable |

A relatively easy way to progress significantly in the field of biosensors for oenology applications is by adapting some of the devices and concepts developed for the biomedical field, which is the major application sector that will continue to drive the biosensors market, expected to reach $31.5 billion by 2024 (as per "Biosensor market" report by Markets and Markets Research Private Ltd, Pune, India, www.marketsandmarkets.com/PressReleases/biosensors.asp, accessed 8 August 2019). Commercial biosensors addressing mainly the biomedical diagnostics and research market are provided, for example, by Abbott Laboratories (Chicago, IL, USA), Abtech Scientific Inc. (Richmond, VA, USA), Bayer Diagnostics (a division of Siemens Healthcare Diagnostics Inc., Los Angeles, CA, USA), Sysmex Corporation (Kobe, Hyogo, Japan), Hoffmann-La Roche Ltd. (Basel, Switzerland), Medtronic Inc. (Dublin, Ireland), Universal Biosensors Inc. (Rowville, Australia), Biacore Life Sciences (now GE Healthcare Life Sciences, Chicago, IL, USA), Affymetrix (now part of ThermoFisher Scientific, Santa Clara, CA, USA), LifeScan (Milpitas, CA, USA), Neogen Life Sciences (Lexington, KY, USA), Bio-Rad laboratories Inc. (Hercules, CA, USA), Applied BioPhysics, (Troy, NY, USA) etc.

In addition, several companies provide screen-printed biosensors for glucose, lactate etc. that can work with hand-held instruments or can be included in FIA systems for wine analysis, e.g., BVT Technologies (Strážek, Czech Republic, www.bvt.cz), Gwent Group(UK, www.gwent.org, now part of Sun Chemical Corporation, Parsippany, New Jersey, USA), Dropsens-Metrohm (Asturias, Spain, www.metrohm.com) etc.

In order to increase the number of commercially available enzymatic, electrochemical biosensors applied in the wine sector some challenges need to be solved, including (1) development of novel/genetically modified redox enzymes with increased stability, specificity and high catalytic activity as biorecognition elements in biosensors, (2) immobilization methodologies to enhance long-term operational stability and (3) efficient electron transfer from the enzyme to the electrode.

### 2.6. Case Study: A Biosensor-Based System for Monitoring Wine Fermentation

While different commercial systems exists for monitoring wine fermentation, they can be complex, expensive and non-adequate for small and medium wine producers. Alternatively, the number of parameters that are monitored are limited. Adding to other efforts for developing flexible, affordable production monitoring systems to assist wine producers, e.g., described in [26–28,49–52] or discussed in Sections 2.1–2.3, we present below a biosensor-based system for monitoring the alcoholic fermentation developed by our group in the frame of a recent Horizon 2020-Eranet, Manunet II project SENS4WINE (2017–2019). The automated monitoring system was intended for medium wine producers, being priced under 15 KEuros and included a sampling unit, a dilution module and an analysis unit, which includes biosensors based on screen-printed electrodes. The system allows the detection of glucose and phenolic compounds along the alcoholic fermentation of white and red wines. In addition, the system includes biosensors for the offline analysis of lysozyme, an antimicrobial, allergenic additive used in wine production (described in [10] and in Section 2.4 above).

To follow the progress of alcoholic fermentation, monitoring the consumption of either glucose or total reducing sugars are easy to implement with commercial electrochemical sensors.

In a first step, an enzymatic sensor based on a screen-printed carbon electrode modified with the electrochemical mediator potassium ferrocyanide and with glucose oxidase (DRP-GLU10) was compared with a catalytic sensor based a gold electrode (DRP-220 AT). The screen-printed electrochemical cell included besides the 4 mm diameter working electrode a silver reference electrode and a counter electrode made of C (DRP-GLU10) or of Au (DRP-220AT). Glucose detection with the enzymatic sensor was performed by chronoanamperometry at −0.1 V, by reading the current intensity at 60 s after applying the potential. The electrodes were held horizontally and 50 μL of sample were applied on the screen-printed electrochemical cell. The calibration curve was linear in the range 0.06–0.6 mM and the current intensity I varied with glucose concentration according to the equation: I (μA) = −2.476 [Glucose] mM-0.082 ($R^2$ = 0.998). By comparison, the non-enzymatic sensor DRP220AT allowed the determination of total reducing sugars (glucose plus fructose) by linear sweep voltammetry, by scanning the potential at a rate of 50 mV s$^{-1}$ in the range from 0.0 V to 1.0 V. The peak current intensity varied linearly with total sugar concentration in the range from 0.05 to 1 mM according to the concentration I (μA) = 77.482 [reducing sugars] mM + 0.899 ($R^2$ = 0.998). The two sensors were applied for the analysis of commercial beverages (red and white must) and fruit acquired in a local market, as well as in the analysis of oenology certified reference grape juice from Titrivin (Blanquefort, France, www.titrivin.com). In a rough approximation, glucose represents half of the amount of reducing sugars. The results in Table 4 emphasize that the GLU10 enzymatic sensor is appropriate for monitoring glucose while DRP220AT is useful for determining the total reducing sugars in must, grape and wine, both sensors providing accurate results that are within 10% difference to those determined in parallel via a standard colorimetric kit (Sigma-Aldrich, Germany). The GLU10 sensor for glucose and the DRP 220AT sensor for total reducing sugars are characterized by a good reproducibility (RSDs between 2.7% and 11.0%).

**Table 4.** Comparative analysis of glucose and total reducing sugars in grape and must with the enzymatic sensor DRP GLU10, the catalytic sensor 220 AT, and a standard enzymatic colorimetric kit.

| Sample | Enzymatic Sensor GLU10 | | Catalytic Sensor 220AT | Colorimetric Kit | | Labeled Value |
|---|---|---|---|---|---|---|
| | Glucose (mM) | Total Sugars [a] (mM) | Reducing Sugars [b] (mM) | Glucose (mM) | Total Sugars [a] (mM) | Reducing Sugars [b] (mM) |
| White must | 455 ± 27 | 910±54 | 998 ± 78 | 496 ± 62 | 992±124 | 944 |
| Red must | 444 ± 29 | 888±58 | 928 ± 85 | 475 ± 32 | 950±64 | 944 |
| Grape juice | 448 ± 13 | 896±26 | 852 ± 86 | 440 ± 14 | 880±28 | 779 |
| White grape | 337 ± 37 | 674±74 | 684 ± 59 | 367 ± 10 | 734±20 | n/a |

[a] estimated value for main reducing sugars (glucose plus fructose) calculated as 2× glucose concentration; [b] reducing sugars: glucose plus fructose; n/a: not applicable; n = 7.

Next, for the selective, online monitoring of glucose during the alcoholic fermentation the enzymatic biosensor GLU10 was inserted in a flow cell and the method was optimized for maximum response and stability. The detection principle is illustrated in Figure 14. In a simple approach, the electrochemical mediator (20 mM $K_4[Fe(CN)_6]$ in 0.1 M TRIS (tris(hydroxymethyl)aminomethane) buffer pH 7.4) was supplied with the electrolyte flow at a rate of 0.5 mL min$^{-1}$. The method was first verified offline using a flow-injection system setup, and then it was implemented within the automated monitoring system and applied for monitoring the alcoholic fermentation of wines. The fermentation of white wines was performed in 500 hL stationary tanks, at 18–19 °C and that of red wines took place simultaneously with the maceration process, in 100 hL rotary tanks. Samples were collected at least twice a day and analyzed in parallel by a standard enzymatic colorimetric test, corresponding to the official method for glucose analysis in wines of OIV. Based on the analysis of 31 white wine and 7 red wine samples, the glucose concentrations measured by the two methods (biosensor and official OIV method using an enzymatic colorimetric kit) are well correlated according to the relation: [glucose] (mM, determined with the kit) = 1.0472 [glucose] (mM, determined with the biosensor) + 1.2952. ($R^2$ = 0.9729). The slope of the regresion line being close to 1, it can be concluded that the biosensor provides equivalent results to the OIV method. The glucose biosensor allowed the main phases characteristic for a classic alcoholic fermentation profile to be observed.

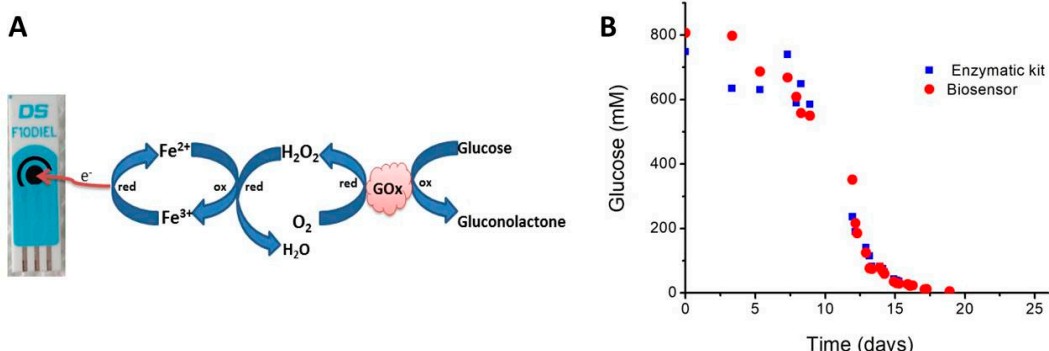

**Figure 14.** (**A**) Principle of the glucose biosensor used for monitoring the alcoholic fermentation. (**B**) Monitoring the consumption of glucose during the alcoholic fermentation of white wines with the biosensor (red) and via parallel tests with a colorimetric kit based on the official method of the Organisation Internationale de la Vigne et du Vin (OIV, blue).

The automated monitoring system (Figure 15A) includes a sample dilution module (Figure 15B) that allows changing the sample dilution factor from 1:1000 at the beginning of the fermentation to 1:50 at the end of the process, to maintain the concentration of diluted samples within the linear range of the biosensor. One glucose biosensor was used continuously for 5 days before being replaced. A periodic calibration consisting of injecting 2 standard solutions of 0.1 and 1 mM glucose, respectively,

two times per day was performed to ensure the system functionality within the optimal operating range. Each standard and sample was injected in triplicate, the standard and electrolyte solutions being prepared daily. The sample concentration was calculated automatically and displayed based on the preceding calibration (Figure 15C).

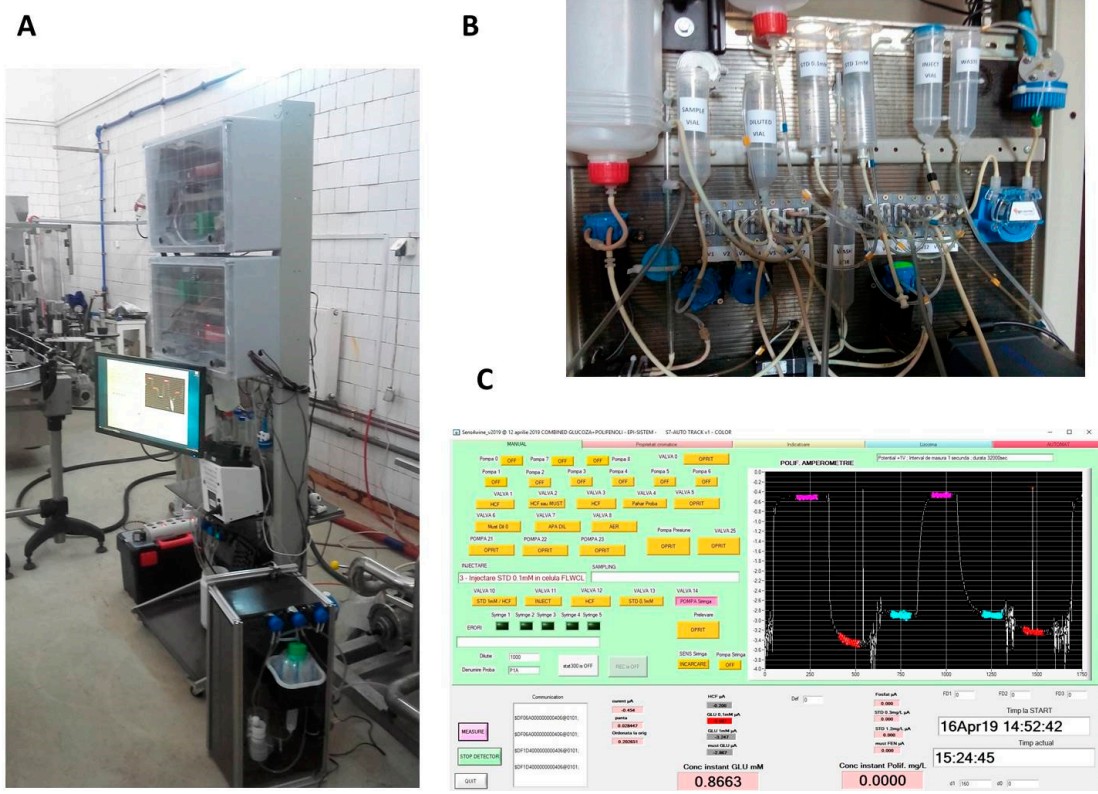

**Figure 15.** (**A**). Automated system based on electrochemical biosensors for monitoring the alcoholic fermentation of wines. (**B**). Details of the sample dilution module. (**C**). User interface display allowing data and system status viewing.

The results obtained show that although the commercial glucose sensors were developed by the manufacturer for single-use applications in the biomedical field, these screen-printed commercial glucose sensors are adequate for monitoring the alcoholic fermentation of wine with a minimum adaptation of the testing protocol and are appropriate to include in automated systems.

Besides glucose, particularly when it comes to red wines, the content of phenolic compounds and wine color are important parameters to control to ensure consistent high quality. In the production of red wines, after grape destemming and crushing, the grape must is kept in contact with the grape skins and seeds to extract compounds responsible for the color (polyphenols), aroma and tannins, contributing to the final organoleptic characteristics. This process, called "maceration", takes places concomitantly with the alcoholic fermentation and the pressing is done immediately afterwards. The maceration-fermentation of black grapes was monitored in fall 2017 and took place in rotary tanks at 25 °C. The progress in the extraction of phenolic compounds, corresponding to the changes in wine color were monitored using the same automated system described above, but using different commercial screen-printed electrode and experimental conditions. The working electrode was a multiwall carbon nanotube-modified carbon electrode (MWCNT DRP110), the electrolyte was a 0.1 M phosphate buffer pH 2.7 at 1 mL min$^{-1}$ and the detection took place also by amperometry at +1.0 V. Gallic acid was used as a standard for calibration and the concentration of total phenolic compounds in the sample was expressed as gallic acid equivalents. An initial verification of the sensor was carried out offline in a flow-injection system, before including the biosensor in the analysis unit of the automated

monitoring system. The FIA tests emphasized a linear range of 0.01–3 mg L$^{-1}$, a detection limit of 9 μg L$^{-1}$, very good stability of the MWCNT-DRP 110 sensor (90% of the initial activity after 5 days of operation) and a good correlation with the content of total phenolic compounds determined via the Folin–Ciocalteu test (R$^2$ = 0.9622). However, the absolute values for the total phenolic compounds represent about half of the magnitude of corresponding concentrations determined for the same samples via the Folin–Ciocalteu method. Thus, at the end of the alcoholic fermentation the red wine produced in the 2017 campaign contained 1.41 g L$^{-1}$ phenolic compounds while the corresponding Folin–Ciocalteu test indicated a value of 2.74 g L$^{-1}$ expressed as gallic acid. This discrepancy, shown also by other authors [4], was attributed to the fact that other easily oxidized compounds besides the phenolics in the sample react with the color reagent in the Folin–Ciocalteu test [120]. Nevertheless, with appropriate calibration and by using a correction factor the sensor can be used to indicate when a certain concentration of total phenolic compounds is reached.

The content of phenolic compounds measured with the MWCNT-modified carbon electrode is, moreover, highly correlated with wine color, e.g., the total color intensity (sum of absorbance at 420 nm, 520 nm and 620 nm). The dynamics of the content of phenolic compounds determined electrochemically with the MWCNT-DRP110 sensor reflects the dynamics of color intensity of wines (Figure 16). Consequently, upon larger-scale verification the MWCNT-DRP110 sensor can be used as an indicator of wine color and alerts can be embedded in the automated system to inform the winemaker when the desired wine color intensity has been reached during the maceration-fermentation, helping to produce batches of consistent characteristics.

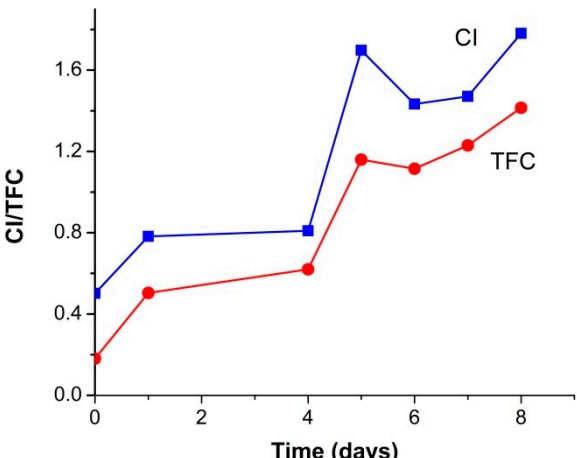

**Figure 16.** Variation of the concentration of total phenolic compounds (TPC, expressed as mg L$^{-1}$ gallic acid) measured with the electrochemical sensor, in comparison with the color intensity index (CI, expressed as absorbance units). Considering all these data, the electrochemical sensor can be used instead of an optical sensor to measure wine color based on previous calibration and by establishing the correlation factor between the current intensity measured with the sensor and wine color.

The results presented above emphasize the usefulness and ease of implementing electrochemical methods based on commercially available sensors in the monitoring of the wine's alcoholic fermentation.

## 3. Conclusions and Perspectives

Wine production is an important economic sector and in wineries investment in new technologies is growing. Despite progress with regard to analytical devices for monitoring the fermentation, biosensors have not convincingly entered the wine analysis market. Many of the older and new technologies for glucose or lactate developed for the biomedical market can, however, be easily used for monitoring AF and MLF with minimal adaptations as shown also in the presented case study. Aside from the reluctance in the wine production field to implement biosensors, there is a constant development effort

for new sensors and multi-analyte detection. Voltamperometric (bio)sensors represent an important part of this research work as the intrinsic advantages of these methods combined with the use of new nanomaterials and composites push the selectivity and sensitivity of detection even further. The potential is high, however: most efforts to tap into this potential are limited to sensors of a new design, tested with small sets of wine samples, with superficial validation and testing in industrial settings.

Although most biosensors for wine analysis are based on enzymes, little of the biosensor development effort in the past 20 years was dedicated to stabilization of enzymes such as glucose oxidase, lactate oxidase or alcohol oxidase, to advance towards commercialization.

In some cases, i.e., for determining the content of total sugars or total polyphenols in wine content, the advances in nanotechnology led to new opportunities to circumvent the need for enzymes as an alternative way to produce robust sensors for industrial processes. The potential of nanomaterials to provide increased stability, sensitivity or biomimetic activity is increasingly being exploited. For example, ceria nanoparticles with oxidase-like activity and dual oxidation states, able to react with polyphenolic compounds were already explored in electrochemical biosensors for measuring the antioxidant activity of wines. Gold electrodes, or electrodes modified with NiO, CuO nanoparticles or composite Pt/MWCNT or Pd/MWCNT nanomaterials have been investigated for the non-enzymatic detection of sugars] etc.

As the electrocatalytic and enzyme-mimicking properties of the nanoparticles depend on their size, zeta potential and operational conditions, significant research efforts to uncover the influence of these factors, tests with large sets of wine samples and steps towards standardizing these new materials are needed as the next step towards commercialization of these new sensor devices.

While the allergen-testing market for wine is dominated by antibody-based analytical devices, with the progress in the field of aptamer it is foreseeable that new, aptamer-based technologies will emerge.

With regard to automated and multi-analyte monitoring, with industrial applications of the Internet of Things (IoT) and artificial intelligence progressing at a fast pace, growing the capacity for large-scale monitoring in wine production is anticipated as well. The monitoring can be easily expanded to multisensory systems including e-tongues and systems combining biosensors with classical physical sensors, provided the issues of biosensor stability and costs are resolved.

The number of voltammetric methods for the determination of polyphenolic compounds and total antioxidant capacity of wines is growing and electrochemical sensors have a good potential to replace the classic methods for measuring the color and total polyphenols and for assessing the oxidation status of wines. Projects focusing on wine production, interlaboratory studies, and the increased use of reference materials will contribute in order to have statistically relevant data, to bring consistency to reporting the research, and to provide the necessary validation that would advance the proposed concepts to commercialization.

**Author Contributions:** A.V.: writing—review and editing, P.E.: review and editing, A.-M.T.; writing—original draft preparation, R.P.; writing—original draft preparation, P.F.-B.; writing—review and editing.

**Funding:** This research was funded by the Romanian Executive Agency for Higher Education, Research, Development and Innovation (UEFISCDI, for AMT, RP, PE and AV) and by the CDTI, Spain (for PF-B), EraNet Manunet II project SENS4WINE.

**Acknowledgments:** 

**Conflicts of Interest:** The authors declare no conflict of interest.

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
