# Peer review of "Progress in Electrochemical (Bio)Sensors for Monitoring Wine Production"

_chemosensors, doi:10.3390/chemosensors7040066_

Round 1

Reviewer 1 Report

The work is well prepared, but in my opinion too little reference to the topic itself.If this is to be a review, then more examples related to the production of wine should be mentioned. I would like to refer to threads connected with Figures 15 and 16 - are these the only examples available?It would be worth to complete it !! besides, a well-planned, well-edited work and perfectly fits into the magazine's trend.

Author Response

Answer: Thank you very much for the kind appreciations and suggestions. To emphasize more strongly the link with the topic and the examples of monitoring wine production we have :

 (1) inserted a line in the introduction to  give some context on the economic importance of wine productions and

(2) revised a paragraph in the beginning of section 2.6, “Case study” to refer to the other examples of biosensor- based monitoring systems for wine production that were discussed or referenced in the previous sections: “ Adding to other efforts for developing flexible, affordable production monitoring systems to assist wine producers, e.g described in [26-28,49-52] or discussed in sections 2.1-2.3, we present below a biosensor based system for monitoring the alcoholic fermentation..” The case study presented is by no means an unique example as we have discussed others in section 2.1 mostly, in addition to the available commercial systems for monitoring wine production presented in section 2.5. The system presented in section 2.6 is intended merely as an illustration of the possibilities offered by electrochemical biosensors for the online monitoring of wine production using commercially available low cost electrodes (being thus in our opinion easier to implement) in a system that was actually tested in real winery conditions.

Reviewer 2 Report

The review is well prepared with a significant number of references cited. I just recommend the authors improve the quality of the images presented in the manuscript.

Author Response

Answer: Thank you very much for the kind appreciations and suggestions. Images were replaced with better quality ones.